# Improving Memory Efficiency for Training KANs via Meta Learning

**Zhangchi Zhao** [1]  **Jun Shu** [1 2]  **Deyu Meng** [1 2]  **Zongben Xu** [1]

## Abstract

Inspired by the Kolmogorov-Arnold representation theorem, KANs offer a novel framework for function approximation by replacing traditional neural network weights with learnable univariate functions. This design demonstrates significant potential as an efficient and interpretable alternative to traditional MLPs. However, KANs are characterized by a substantially larger number of trainable parameters, leading to challenges in memory efficiency and higher training costs compared to MLPs. To address this limitation, we propose to generate weights for KANs via a smaller meta-learner, called MetaKANs. By training KANs and MetaKANs in an end-to-end differentiable manner, MetaKANs achieve comparable or even superior performance while significantly reducing the number of trainable parameters and maintaining promising interpretability. Extensive experiments on diverse benchmark tasks, including symbolic regression, partial differential equation solving, and image classification, demonstrate the effectiveness of MetaKANs in improving parameter efficiency and memory usage. The proposed method provides an alternative technique for training KANs, that allows for greater scalability and extensibility, and narrows the training cost gap with MLPs stated in the original paper of KANs. Our code is available at https://github.com/Murphyzc/MetaKAN.

## 1. Introduction

With the rapid development of deep learning, artificial intelligence (AI) has achieved remarkable progress in various fields (Brown et al., 2020; Jumper et al., 2021). This progress is largely due to advancements of various neural network architectures, ranging from MLP (Cybenko, 1989) to CNN (LeCun et al., 1989), RNN (Elman, 1990), LSTM (Hochreiter & Schmidhuber, 1997), and Transformer (Brown et al., 2020) architectures. While these architectures generally use fixed activation functions, which limit their interpretability and flexiblity.

Recently, Kolmogorov-Arnold Networks (KANs)(Liu et al., 2024), inspired by the Kolmogorov-Arnold representation theorem(Kolmogorov, 1957), introduced a novel neural architecture. Rather than employing traditional fixed activation functions, KANs utilize learnable, spline-parametrized univariate functions on edges, enabling a more flexible and interpretable framework for function approximation. Recent research on KANs has focused on modifying the B-spline functions to improve their performance. For instance, Chebyshev KAN (SS et al., 2024), GramKAN, and KALN used Chebyshev polynomials, Gram polynomials, and Legendre polynomials, respectively, to enhance the non-linear function approximation capabilities. Similarly, (Bozorgasl & Chen, 2024) used wavelet basis functions to improve the model's ability of capturing both high- and low-frequency components of data. Additionally, (Yang & Wang, 2024) and (Li, 2024) replaced B-splines with rational polynomials and radial basis functions to improve the computational speed of KANs. To enhance KANs' capability in image classification tasks, (Bodner et al., 2024) proposed ConvKAN, while (Drokin, 2024) extended ConvKAN to more convolutional structures. Although these KAN variants improve approximation and computational efficiency, the presence of learnable activation functions leads to a large number of trainable parameters. This design significantly impacts memory efficiency and increases training costs, posing challenges for addressing larger datasets and complex tasks.

To address these challenges, we propose MetaKANs, a memory-efficient framework that improves training efficiency for KANs through meta learning. Our approach builds on the key observation: KANs decompose high-dimensional functions into dimension-wise univariate functions (learnable, spline-parametrized activation functions), while these univariate functions follow a common functional class $\mathcal{F}$, in which they share the same trainable parameters setting rule. Inspired by current meta learning (Kong et al., 2020; Shu et al., 2023) and in-context learning (Brown et al., 2020; Garg et al., 2022) techniques, we propose to

[1]School of Mathematics and Statistics, Ministry of Education Key Lab of Intelligent Networks and Network Security, Xi'an Jiaotong University [2]Pengcheng Laboratory. Correspondence to: Jun Shu <junshu@mail.xjtu.edu.cn>.

*Proceedings of the $42^{nd}$ International Conference on Machine Learning*, Vancouver, Canada. PMLR 267, 2025. Copyright 2025 by the author(s).

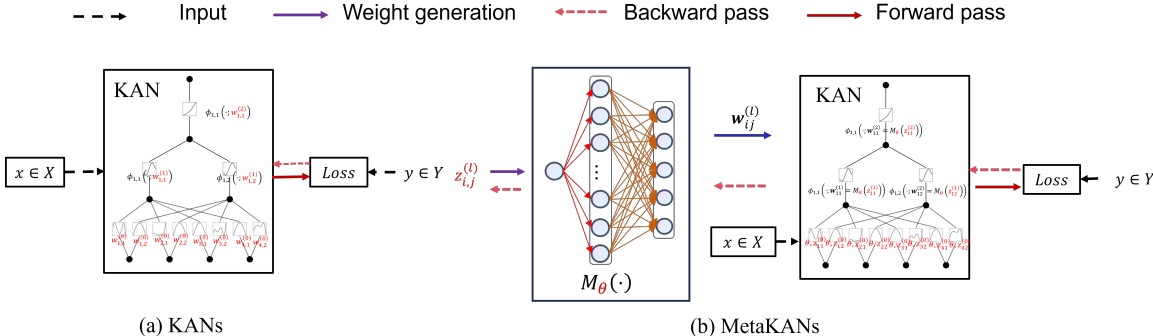

*Figure 1.* Overview of the architectures for KANs and MetaKANs. The connections marked red means trainable. The trainable parameters are $\mathbf{w}_{i,j}^{(l)}$ for KANs and $\theta, z_{i,j}^{(l)}$ for MetaKANs.

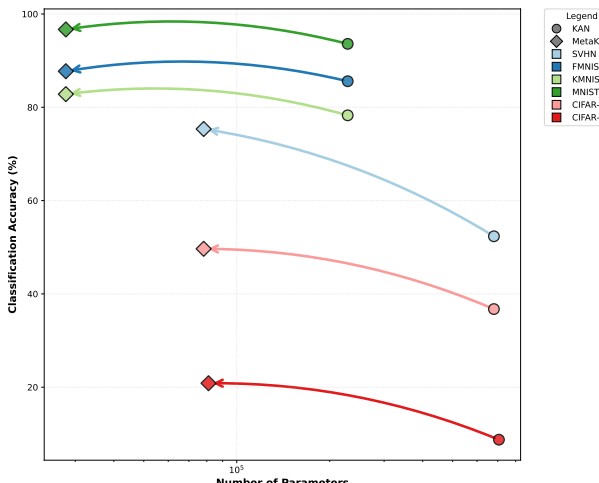

*Figure 2.* Parameter-accuracy tradeoff comparison between KANs and MetaKANs across six benchmark datasets. Marker color indicates dataset, shape represents model type (circle: KANs, square: MetaKANs). MetaKANs demonstrates superior memory efficiency while maintaining competitive performance. The arrow indicates that MetaKANs not only reduce the number of parameters but also maintain performance.

use a smaller meta-learner to simulate the shared parameter generation rule, and thus the large number of trainable parameters for activation functions reduce to the parameters of the smaller meta-learner. Specifically, the parameters of MetaKANs consists of two parts: (1) learnable input variable (i.e., prompts) serving as the unique identifier for each activation function and (2) the parameters of meta-learner learning the shared weight setting rule for dimension-wise univariate functions. The diagram is shown in Figure 1.

In a nutshell, we make the following contributions:

(1) We propose a novel memory efficiency training method for KANs, called MetaKANs, which generates weights of KANs via a smaller meta-learner instead of directly optimizing parameters of KANs. As shown in Figure 2, proposed method achieves comparable or even better performance with only 1/3 to 1/9 parameters of the number of trainable parameters for KANs.

(2) Proposed method is model-agnostic, which could be applied to various KAN variants. The memory efficiency improvements upon FastKAN (Li, 2024), ConvKAN (Bodner et al., 2024), and WavKAN (Bozorgasl & Chen, 2024) demonstrate the universal applicability for MetaKANs.

(3) We experimentally demonstrate that MetaKANs improve training efficiency in terms of reducing memory load across various benchmarks, including symbolic regression, solving PDEs, and image classification. Experimental results substantiate the capability of MetaKANs on enhancing the scalability and extensibility of KANs.

## 2. Related Work

### 2.1. Kolmogorov-Arnold Networks (KANs)

KANs (Liu et al., 2024) introduce a novel network structure by replacing fixed activation functions in traditional MLP networks with learnable activation functions. This provides a more flexible and interpretable framework for function approximation tasks. Recent studies on KANs can be broadly categorized into three research directions: parameterization strategies of basis functions, domain-specific applications, and theoretical foundations.

To enhance expressivity and computational efficiency, diverse basis functions have been explored beyond the original B-spline formulation. These include polynomial bases such as Chebyshev, Legendre, and Gram polynomials (SS et al., 2024), rational functions (Aghaei, 2024), radial basis functions (Li, 2024), wavelets (Bozorgasl & Chen, 2024). Besides, KANs have been integrated into diverse architectures and tasks. In computer vision, ConvKAN (Bodner et al., 2024) and KAT (Yang & Wang, 2024) enhance CNNs and Transformers, respectively. For time-series, frequency-aware models like TIMEKAN (Huang et al., 2025) and memory-augmented variants (Genet & Inzirillo, 2024) improve long-term forecasting. U-KAN (Li et al., 2024) applies KANs to U-Net for medical imaging, while KAN-ODEs (Koenig et al., 2024) model dynamical systems. In graphs, KA-GNNs (Bresson et al., 2024) and Kolmogorov-

Arnold Attention (Fang et al., 2025) boost expressive capacity. Other extensions include continual learning classifiers (Hu et al., 2025) and generative modeling via learned Schrödinger bridges (Qiu et al., 2025). Theoretical work has begun to formalize KANs' capacity and generalization properties. (Zhang & Zhou, 2024) established generalization bounds, quantifying KAN complexity based on activation properties. While expressiveness studies show that KANs can outperform standard MLPs under large-grid regimes and exhibit weaker spectral bias—suggesting better modeling of high-frequency components (Wang et al., 2024).

Although these developments have advanced KANs significantly, the presence of learnable activation functions results in a large number of trainable parameters and challenges in memory efficiency than MLP network, posing challenges for addressing larger datasets and complex tasks. Unlike these prior works, our proposed MetaKANs leverage a meta-learner and learnable prompts corresponding to activation functions, achieving a trainable parameter count comparable to that of MLPs with similar structures.

## 2.2. HyperNetworks

HyperNetworks use a smaller auxiliary network to generate the weights of a larger main network, offering a mechanism to reduce trainable parameter counts while maintaining competitive performance. First introduced in (Ha et al., 2017), hypernetworks achieved comparable accuracy to typical CNNs and RNNs with significantly fewer parameters, laying the foundation for a broad range of follow-up research. Recently, Mahabadi et al. (2021) applied hypernetworks to Transformers for efficient fine-tuning across multiple tasks

In meta-learning, (Zhao et al., 2020) reformulated the template-sharing mechanism as a hypernetwork guided by learnable input vectors. (Navon et al., 2021) extended this idea to multi-objective settings, using task embeddings to condition a shared hypernetwork, allowing task-specific weight generation without duplicating full networks. Hypernetworks have also been tailored to specific domains. For instance, (Alaluf et al., 2022) adapted the concept to StyleGAN, enabling efficient, real-time image editing by modulating generator weights with compact hypernetworks.

Prior work typically employs heuristic parameter generation strategies, such as using shared hypernetworks for entire layers (Ha et al., 2017) or task-agnostic templates (Savarese & Maire, 2019). In contrast, propsoed MetaKANs is grounded in the understanding of working mechanism of KANs, in which the weights learning could be regarding as the multi-task basis representation learning from a shared functional family. Instead of heuristically predicting weights through hypernetworks, we explicitly model the shared weight generation rules through a meta-learner, where the learnable prompts as the identifiers for activation functions.

## 3. Methods

### 3.1. Preliminary

#### 3.1.1. KOLMOGOROV-ARNOLD REPRESENTATION THEOREM

The KA representation theorem (Kolmogorov, 1957) states that any multivariate continuous function can be decomposed into a finite sum of univariate functions. Specifically, for a continuous function on $f : [0,1]^n \to \mathbb{R}$, there exist continuous univariate functions $\phi_q^{\text{out}} : \mathbb{R} \to \mathbb{R}$ and $\phi_{q,p}^{\text{in}} : [0,1] \to \mathbb{R}$ such that:

$$f(\mathbf{x}) = f(x_1, ..., x_n) = \sum_{q=1}^{2n+1} \phi_q^{\text{out}} \left( \sum_{p=1}^{n} \phi_{q,p}^{\text{in}}(x_p) \right). \quad (1)$$

This theorem could be formulated as follows:

$$f(\mathbf{x}) = \Phi_{\text{out}} \circ \Phi_{\text{in}} \circ \mathbf{x},$$

where $\Phi_{in}$ and $\Phi_{\text{out}}$ are defined as:

$$\Phi_{in} = \begin{bmatrix} \phi_{1,1}^{\text{in}}(\cdot) & \cdots & \phi_{1,n}^{\text{in}}(\cdot) \\ \vdots & \ddots & \vdots \\ \phi_{2n+1,1}^{\text{in}}(\cdot) & \cdots & \phi_{2n+1,n}^{\text{in}}(\cdot) \end{bmatrix},$$
$$\Phi_{\text{out}} = \begin{bmatrix} \phi_1^{\text{out}}(\cdot) & \cdots & \phi_{2n+1}^{\text{out}}(\cdot) \end{bmatrix}.$$

#### 3.1.2. KOLMOGOROV-ARNOLD NETWORKS

KANs implement the KA representation theorem using a parameterized neural network. Each activation function in KANs is represented by a basis function and its associated parameters. In practice, Liu et al. (2024) explicitly parameterize each univariate function $\phi(\cdot)$ as a B-spline curve, with learnable coefficients of local B-spline basis functions,

$$\phi(t; \mathbf{w}) = w_b \texttt{SiLU}(t) + \sum_{i=1}^{G+k} c_i B_i(t), \quad (2)$$

where $\texttt{SiLU}(t) = \frac{t}{1+e^{-t}}$, $\mathbf{w} = [w_b, c_1, ..., c_{G+k}]^\top$, and $\{B_i(t), i \in [G + k]\}$ is the B-spline basis functions determined by the grid size $G$, spline order $k$. The parameters $w_b$ controls the overall magnitude of the activation function, while $c_i$ are trainable coefficients. We denote $\mathbf{B}(t) = [\texttt{SiLU}(t), B_1(t), ..., B_{G+k}(t)]^T$, so the learnable univariate activation function can be written as:

$$\phi(t; \mathbf{w}) = \mathbf{w}^\top \mathbf{B}(t). \quad (3)$$

The general KANs consist of $L$ layers, with the structure $[n_0, n_1, \ldots, n_L]$, where $n_l$ denotes the number of nodes in the $l$-th layer. The outputs of $l$-th KAN layer is defined as

$$\mathbf{x}^{(l+1)} = \left[ \sum_{i=1}^{n_l} \phi_{i,1}(x_i^{(l)}; \mathbf{w}_{i,1}^{(l)}), \ldots, \sum_{i=1}^{n_l} \phi_{i,n_{l+1}}(x_i^{(l)}; \mathbf{w}_{i,n_{l+1}}^{(l)}) \right],$$
$$= \Phi_{\mathbf{W}^{(l)}}^{(l)} \circ \mathbf{x}^{(l)}, \quad (4)$$

*Table 1.* Comparison of parameter number among different models with the architecture $[n_0, n_1, ..., n_L]$.

| ARCHITECTURES | # PARAMS. |
|---|---|
| MLP | $\sum_{l=0}^{L-1}(n_l \times n_{l+1})$ |
| KAN | $\sum_{l=0}^{L-1}(n_l \times n_{l+1}) \times (G+k+1)$ |
| METAKAN | $\sum_{l=0}^{L-1}(n_l \times n_{l+1}) + C \times (d_{hidden}+1) \times (G+k+1)$ |
| FASTKAN | $\sum_{l=0}^{L-1}(n_l \times n_{l+1}) \times c$ |
| METAFASTKAN | $\sum_{l=0}^{L-1}(n_l \times n_{l+1}) + (d_{hidden}+1) \times c$ |
| WAVKAN | $\sum_{l=0}^{L-1}(n_l \times n_{l+1}) \times 3$ |
| METAWAVKAN | $\sum_{l=0}^{L-1}(n_l \times n_{l+1}) + (d_{hidden}+1) \times 3$ |

where $\mathbf{x}^{(l)} \in \mathbb{R}^{n_l}$ is the input of $l$-th layer, and $\Phi_{\mathbf{W}^{(l)}}^{(l)}$ is an activation matrix consisting of $n_{l+1} \times n_l$ activation functions $\phi_{i,j}(\cdot; \mathbf{w}_{i,j}^{(l)})$, which is given by:

$$\Phi_{\mathbf{W}^{(l)}}^{(l)} = \begin{bmatrix} \phi_{1,1}(\cdot; \mathbf{w}_{1,1}^{(l)}) & \cdots & \phi_{1,n_l}(\cdot; \mathbf{w}_{1,n_l}^{(l)}) \\ \vdots & \ddots & \vdots \\ \phi_{n_{l+1},1}(\cdot; \mathbf{w}_{n_{l+1},1}^{(l)}) & \cdots & \phi_{n_{l+1},n_l}(\cdot; \mathbf{w}_{n_{l+1},n_l}^{(l)}) \end{bmatrix}.$$

The $l$-th KAN layer is parameterized by $\mathbf{W}^{(l)} = \{\mathbf{w}_\alpha^{(l)}, \alpha \in \mathcal{I}_l\}$. Here we denote $[n] = \{1, ..., n\}$ and the index set for the activation functions as $\mathcal{I}_l = [n_l] \times [n_{l+1}]$ for $l \in [L] - 1$. Thus KAN with structure $[n_0, n_1, \ldots, n_L]$ has a set of parameters denoted by:

$$\mathcal{W} = \bigcup_{l=0}^{L-1} \mathbf{W}^{(l)} = \bigcup_{l=0}^{L-1} \left\{ \mathbf{w}_\alpha^{(l)} \in \mathbb{R}^{G+k+1} \mid \alpha \in \mathcal{I}_l \right\}. \quad (5)$$

Given an input vector $\mathbf{x} \in \mathbb{R}^{n_0}$, the output of the KAN is

$$\text{KAN}(\mathbf{x}; \mathcal{W}) = \Phi_{\mathbf{W}^{(L-1)}}^{(L-1)} \circ \cdots \Phi_{\mathbf{W}^{(1)}}^{(1)} \circ \Phi_{\mathbf{W}^{(0)}}^{(0)}(\mathbf{x}). \quad (6)$$

It can be seen that the KA representation theorem in Eq. (1) corresponds to a 2-layer KAN with shape $[n, 2n + 1, 1]$.

KANs have been demonstrated superiority in fucntion fitting than MLP network (Liu et al., 2024). Suppose the target function is $f(\mathbf{x})$, then the learning objective can be formulated as

$$\mathcal{W}^* = \arg\min_{\mathcal{W}} \mathbb{E}_{\mathbf{x} \sim P(\mathbf{x})} \ell(\text{KAN}(\mathbf{x}; \mathcal{W}), f(\mathbf{x})), \quad (7)$$

where $\ell$ denoted the loss function, and we set MSE for regression and cross-entropy loss for classification.

### 3.1.3. MEMORY INEFFICIENCY

KANs differ from MLPs by placing learnable coefficients in each activation function, while MLPs place fixed activation function in the architecture design. This could increase the number of parameters compared with MLPs, especially as the growth of network's width and depth.

From the above, it can be concluded that for KANs with the structure $[n_0, n_1, \ldots, n_L]$, a B-spline function of order $k$ with $G$ grid points require parameters $\mathcal{W}$ as presented in

Eq.(5). These parameters collectively account for the total number of learnable parameters in KANs, which can be expressed by $\sum_{l=0}^{L-1}(n_l \times n_{l+1}) \times (G+k+1)$.

For comparison, the forward computation in an MLP can be expressed as follows

$$\text{MLP}(\mathbf{x}) = \left( \mathbf{W}^{(L-1)} \circ \sigma \circ \cdots \mathbf{W}^{(1)} \circ \sigma \circ \mathbf{W}^{(0)} \right) \mathbf{x}.$$

where $\mathbf{W}^{(l)} \in \mathbb{R}^{n_{l+1} \times n_l}$ represents the linear transformation, and $\sigma$ denotes the nonlinear fixed activation function. Thus an MLP with the same structure requires only $\sum_{l=0}^{L-1}(n_l \times n_{l+1})$ parameters. This shows that the parameter count for a KAN is approximately $(G + k) \times$ larger than that of an MLP (refer to Table 1). Such an increase in parameter count introduces challenges for KANs, including greater memory usage and potential scalability issues.

### 3.2. Improving Memory Efficiency via Meta Learning

#### 3.2.1. UNDERSTANDING WORKING MECHANISM OF 2-LAYER KANS

Consider an $n$-dimensional input $\mathbf{x} = (x_1, \ldots, x_n)$ and a target function $f(\mathbf{x})$ represented by the Eq. (1) according the KA representation theorem. Thus $f(\mathbf{x})$ could be approximated by 2-Layer KANs with shape $[n, 2n + 1, 1]$

$$\begin{aligned} f(\mathbf{x}) &= \sum_{j=1}^{2n+1} \phi_j^{(1)} \left( \sum_{i=1}^{n} \phi_{i,j}^{(0)}(x_i) \right) \\ &\approx \sum_{j=1}^{2n+1} \phi_j^{(1)} \left( \sum_{i=1}^{n} \phi_{i,j}^{(0)}(x_i; \mathbf{w}_{i,j}^{(0)}); \mathbf{w}_j^{(1)} \right) \\ &= \text{KAN}(\mathbf{x}; \mathcal{W}). \end{aligned}$$

Here we could understand the learning of weights $\mathcal{W} = \mathbf{W}^{(0)} \cup \mathbf{W}^{(1)}$ as a multi-task learning problem. Specifically, for the 1-layer weights $\mathbf{W}^{(0)}$, the weights learning is to learn $n \times (2n + 1)$ regression tasks belonging to the shared function class $\mathcal{F}_1 = \{f | f(x^{(0)}) = \mathbf{w}^\top \mathbf{B}(x^{(0)}), \mathbf{w} \in \mathbb{R}^{G+k+1}\}$; and the learning of 2-layer weights $\mathbf{W}^{(1)}$ is to learn $(2n + 1) \times 1$ regression tasks belonging to the shared function class $\mathcal{F}_2 = \{f | f(x^{(1)}) = \mathbf{w}^\top \mathbf{B}(x^{(1)}), \mathbf{w} \in \mathbb{R}^{G+k+1}\}$, where $\mathbf{B}(\cdot)$ is defined in Section 3.1.2. In fact, each regres-

sion task could be expressed by

$$M : \mathcal{T} \to \mathbb{R}^{G+k+1}, \tag{8}$$

where $\mathcal{T}$ denotes the task information to identify which activation function it is, and the output $\mathbf{w} \in \mathbb{R}^{G+k+1}$ corresponds to the learnable coefficients for the activation function. The regular learning of KANs in Eq.(7) overlooks the underlying common weights learning rule, and optimize the weights in a brute-force way. This learning strategy significantly impacts the computation and memory efficiency of KANs, posing challenges of KANs for addressing larger datasets and complex tasks compared with MLP network.

### 3.2.2. PROPOSED METAKANS FRAMEWORK

To improve the memory efficiency of KANs, the key insight is to delicately employ the commom weights learning rules in Eq.(8). To this aim, we proposed a Meta-KANs framework as shown in Figure 1 to directly predict weights of KANs inspired by current advances on meta-learning (Kong et al., 2020; Shu et al., 2018; 2023) and in-context learning (Brown et al., 2020; Xu et al., 2024). Specifically, we design a meta-learner to parameterize the weights learning rules in Eq.(8) as follows

$$M_\theta : \mathbb{R} \to \mathbb{R}^{G+k+1}, \quad z \mapsto \mathbf{w}. \tag{9}$$

Here we use learnable task prompt $z \in \mathbb{R}$ to identify which activation function it is as popular studied in current large language models (Brown et al., 2020), and $M_\theta$ is formulated as a two-layer MLP with hidden dim equals to $d_{\text{hidden}}$. This choice is motivated by the MLP's capability as a universal function approximator, as used in previous meta-learning methods (Ha et al., 2017; von Oswald et al., 2020; Shu et al., 2019) and its simplicity and computational efficiency.

Based on this formulation, the learnable activation function in Eq. (3) now becomes:

$$\phi\left(t; z, \theta\right) = M_\theta(z)^\top \mathbf{B}(t), \tag{10}$$

and then the 2-layer KAN in Eq. (8) now becomes:

$$\text{MetaKAN}(\mathbf{x}; \mathcal{Z}, \theta) = \sum_{j=1}^{2n+1} \phi_j^{(1)} \left( \sum_{i=1}^{n} \phi_{i,j}^{(0)} \left( x_i; M_\theta(z_{i,j}^{(0)}) \right); M_\theta(z_j^{(1)}) \right).$$

Note that the learnable parameters $\mathbf{w}$ in activation function of KANs are converted into the parameters $(\mathcal{Z}, \theta)$ of the meta-learner, where $\mathcal{Z}$ is defined as (see Sec.4.4)

$$\mathcal{Z} = \bigcup_{l=0}^{L-1} \left\{ z_\alpha^{(l)} \in \mathbb{R} \mid \alpha \in \mathcal{I}_l \right\}, \mathcal{I}_l = [n_l] \times [n_{l+1}]. \tag{11}$$

This definition corresponds to the general case of a $L$-layer KAN, and the previously discussed case with $L = 2$ is a

---

**Algorithm 1** The MetaKANs Algo. for Shallow KANs

**Input:** Training dataset $\mathcal{D}^{tr}$, batch size $n$, max iterations $T$, KAN and MetaKAN models,
**Output:** Updated meta-learner parameters $\theta^{(T)}$ and prompts $\mathcal{Z}^{(T)}$
Initialize meta-learner parameters $\theta^{(0)}$ and learnable prompts $\mathcal{Z}^{(0)}$
**for** $t = 0$ **to** $T - 1$ **do**
    Sample a mini-batch $\{\mathbf{x}_b, \mathbf{y}_b\}$ from $\mathcal{D}^{tr}$ of size $n$
    Compute the output of MetaKAN using Eq. (3.2.2)
    Compute training loss with Eq. (12)
    Update $\theta^{(t+1)}$ and $\mathcal{Z}^{(t+1)}$ via gradient descent
**end for**

---

special instance. Thus the number of trainable parameters of MetaKANs is

$$|\mathcal{Z}| + |\theta| = \sum_{l=0}^{L-1} (n_l \times n_{l+1}) + (d_{hidden} + 1) \times (G + k + 1).$$

To obtain the meta-learner, we can optimize the following objective function derived from Eq.(7):

$$\mathcal{Z}^\star, \theta^\star = \arg \min_{\mathcal{Z}, \theta} \mathbb{E}_{\mathbf{x} \sim P(\mathbf{x})} \left[ \ell(\text{MetaKAN}(\mathbf{x}; \mathcal{Z}, \theta) - f(\mathbf{x})) \right].$$

$$\tag{12}$$

During the training process, both the parameters of the meta-learner $\theta$ and the learnable prompts $\mathcal{Z}$ are updated simultaneously by gradient descent algorithm. The overall training algorithm for MetaKANs is formulated in Algorithm.1.

### 3.2.3. SCALING TO DEEP KANS

Based on the analysis in Section 3.2.1, we could further understand the weights learning for a multi-layer KAN defined in Eq. (6). Specifically, for the $l$-layer weights $\mathbf{W}^{(l)}$, the weights learning is to learn $n_l \times n_{l+1}$ regression tasks belonging to the shared function class $\mathcal{F}_l = \{f | f(x^{(l)}) = \mathbf{w}^\top \mathbf{B}(x^{(l)}), \mathbf{w} \in \mathbb{R}^{G+k+1}\}$. In other words, the weights learning of deep layer of KANs is akin to that of the shallow layer, while the input task information is shift and evolutionary across different layers. This mismatch task distribution makes the weighting learning rule in Eq.(8) different across layers. Though we introduce the learnable prompts $\mathcal{Z}$ as input information for each task, it is hard to mine the proper layer information use a shared meta-learner, and thus bring inferior performance as demonstrated in Table 9.

To address this issue, a straightforward way is to assign different meta-learners for every layer. This design allows the meta-learner $M_\theta^{(l)}$ for $l$-th layer to effectively capture the layer-dependent task information, which could potentially help extract proper weights learning rules for different layers. However, such setup would introduce significant parameter overhead because that the total parameters are scaling with the number of layers $L$, which tends to further raise their memory inefficiency for deep KANs with many layers.

**Algorithm 2** Clusters Determination Algo. for Deep KANs

1: **Input:** Output-channel sizes $\mathbf{n} = [n_1, \ldots, n_{L-1}]$; number of clusters $C$
2: **Output:** Intervals $\left\{ (l_{\text{start}}^c, l_{\text{end}}^c) \right\}_{c=1}^C$
3: $labels \leftarrow \text{KMEANS}(\mathbf{n}, C)$
4: **for** $c = 1$ **to** $C$ **do**
5: $\quad L_c \leftarrow \{ l \mid labels[l] = c \}$
6: $\quad l_c^{\text{start}} \leftarrow \min L_c, \quad l_c^{\text{end}} \leftarrow \max L_c$
7: **end for**
8: **return** $\left\{ (l_c^{\text{start}}, l_c^{\text{end}}) \right\}_{c=1}^C$

To make a balance between expressive capacity and memory efficiency, we extend the MetaKANs framework by partitioning the deep KAN's $L$ layers into $C$ distinct clusters, $\{L_1, \ldots, L_C\}$, each assigned a dedicated meta-learner $M_{(c)}$. This partitioning is guided by the input and output channels of the layers, a strategy that typically groups consecutive layers $L_c = [l_c^{\text{start}}, l_c^{\text{end}}]$ which exhibit similar channel dimension, as illustrated for a ConvKAN in Sec. C.4. The detailed algorithm please refer to Algorithm 2.

Based on this novel formulation, the parameters in Eq. (5) are generated with proposed MetaKANs:

$$\mathbf{w}_\alpha^{(l)} = M_{\theta_{(c)}} \left( z_\alpha^{(l)} \right), \quad \forall l \in L_c, \alpha \in \mathcal{I}_l \qquad (13)$$

where $\theta_{(c)}$ represents the cluster-specific meta-learener parameters. And we will validate this design in Sec. C.4. Then the output of the $l$-th KAN layer in Eq. (4) is given by:

$$
\begin{aligned}
\mathbf{x}^{(l+1)} &= \Phi_{\mathbf{z}^{(l)}, \theta_{(c)}}^{(l)} \circ \mathbf{x}^{(l)} \\
&= \left[ \sum_{i=1}^{n_l} \phi_{i,1}(x_i^{(l)}; M_{\theta_{(c)}}(z_{i,1}^{(l)})), \ldots, \right. \\
&\quad \left. \sum_{i=1}^{n_l} \phi_{i,n_{l+1}}(x_i^{(l)}; M_{\theta_{(c)}}(z_{i,n_{l+1}}^{(l)})) \right], l \in L_c.
\end{aligned}
\qquad (14)
$$

The forward procedure of MetaKANs now becomes:

$$\text{MetaKAN}(\mathbf{x}; \mathcal{Z}, \Theta) = \Phi_{\mathbf{z}^{(L-1)}, \theta_{(C)}}^{(L-1)} \circ \cdots \circ \Phi_{\mathbf{z}^{(0)}, \theta_{(1)}}^{(0)}(\mathbf{x}),$$

where $\Theta = \{\theta_{(c)}\}_{c=1}^C$, and the objective function is

$$
\begin{aligned}
\mathcal{Z}^\star, \Theta^\star &= \\
\arg &\min_{\mathcal{Z}, \Theta} \mathbb{E}_\mathbf{x} \left[ \left( \text{MetaKAN}(\mathbf{x}; \mathcal{Z}, \Theta) - f(\mathbf{x}) \right)^2 \right].
\end{aligned}
\qquad (15)
$$

The overall training algorithm for MetaKANs to predict weights of deep KANs is formulated in Algorithm.3.

**Memory efficiency analysis.** The number of learnable parameters in MetaKANs for deep KANs with the structure $[n_0, n_1, ..., n_L]$ is given by:

$$|\mathcal{Z}| + |\Theta| = \sum_{l=0}^{L-1} (n_l \times n_{l+1}) + C \times (d_{hidden} + 1) \times (G + k + 1).$$

**Algorithm 3** The MetaKANs Algo. for Deep KANs

1: **Input:** Training dataset $\mathcal{D}^{tr}$; KANs with $L$ layers architecture and MetaKANs; Layer clustering $\{L_c\}_{c=1}^C$ obtained using Algorithm 2, where $L_c = [l_c^{\text{start}}, l_c^{\text{end}}]$; Batch size $n$; Max iterations $T$
2: **Output:** Updated meta-learner parameters $\Theta^{(T)} = \left\{ \theta_{(c)}^{(T)} \right\}_{c=1}^C$ and prompts $\mathcal{Z}^{(T)}$
3: Initialize meta-learner parameters $\Theta^{(0)} = \left\{ \theta_{(c)}^{(0)} \right\}_{c=1}^C$ and learnable prompts $\mathcal{Z}^{(0)}$
4: **for** $t = 0$ **to** $T - 1$ **do**
5: $\quad$ Sample a mini-batch $\{\mathbf{x}_b, \mathbf{y}_b\}$ from $\mathcal{D}^{tr}$ of size $n$
6: $\quad$ **for** $l = 0$ **to** $L - 1$ **do**
7: $\quad\quad$ Generate weights for $l$-th layer using Eq.(13)
8: $\quad\quad$ Propagate through $l$-th KAN layer via Eq. (14)
9: $\quad$ **end for**
10: $\quad$ Compute training loss with Eq. (15)
11: $\quad$ Update $\Theta^{(t+1)}$ and $\mathcal{Z}^{(t+1)}$ via gradient descent
12: **end for**

Compared with case for shallow KANs, the increased number of learnable parameters for deep KANs attributes to the additional number of meta-learners ($C = 1$ degenerates to the case discussed in Section 3.2.2). Their parameters $C \times (d_{hidden} + 1) \times (G + k + 1)$ may remain constant due to the fixed architecture of meta-learners $M_{\theta_{(c)}}(z), c \in [C]$. While the term $\sum_{l=0}^{L-1} (n_l \times n_{l+1})$, which accounts for the learnable prompts $\mathcal{Z}$, will grow with the size of the KANs's architecture. Then the total parameter count of MetaKANs is approximately by

$$|\mathcal{Z}| + |\Theta| \approx \sum_{l=0}^{L-1} (n_l \times n_{l+1}),$$

which is equivalent to that of a standard MLP network with the same structure. The detailed comparison of the number of trainable parameter among KAN, MLP and MetaKAN is shown in Table 1. And thus the proposed MetaKANs framework is potentially promising in improving memory efficiency of training KANs, and reduced memory cost approximating to MLP network. In Section 4, we experimentally demonstrate that proposed MetaKANs could achieve competitive performance than orginal KANs.

### 3.3. Extension to Other KANs Variants

It is worth emphasizing that proposed MetaKANs framework is model-agnostic, which is capable of applying to recent KANs variants, e.g., WavKAN(Bozorgasl & Chen, 2024), FastKAN(Li, 2024) and ConvKAN(Drokin, 2024), etc. To illustrate this, we propose MetaKAN variants that apply our meta-learning framework (Section 3.2) to these architectures including MetaWavKAN, MetaFastKAN, and MetaConvKAN. Detailed descriptions and technical details are provided in Appendix A. And we empirically verify the

Table 2. Performance Comparison Between KANs and MetaKANs on Feynman Dataset with Different Grids.

| Name | Structure | G = 5 | | | | G = 20 | | | |
|---|---|---|---|---|---|---|---|---|---|
| | | KANs | | MetaKANs | | KANs | | MetaKANs | |
| | | MSE | # Param | MSE | # Param | MSE | # Param | MSE | # Param |
| I.6.20A | [1,2,1,1] | $5.94 \times 10^{-4}$ | **58** | **$3.88 \times 10^{-4}$** | 218 | $1.01 \times 10^{-2}$ | **134** | **$5.48 \times 10^{-3}$** | 653 |
| I.6.20 | [2,2,1,1] | $6.44 \times 10^{-3}$ | **76** | **$2.67 \times 10^{-3}$** | 210 | $6.00 \times 10^{-3}$ | **193** | **$4.15 \times 10^{-4}$** | 439 |
| I.6.20B | [3,5,5,5,1] | $4.16 \times 10^{-3}$ | **646** | **$3.68 \times 10^{-3}$** | 1083 | $3.79 \times 10^{-2}$ | 1976 | $3.04 \times 10^{-2}$ | **1,798** |
| I.8.4 | [4,5,5,1] | $1.30 \times 10^{-2}$ | 979 | **$3.76 \times 10^{-3}$** | 461 | $1.16 \times 10^{-1}$ | 1,461 | **$8.33 \times 10^{-2}$** | 1,994 |
| I.9.18 | [6,5,5,5,1] | $2.39 \times 10^{-3}$ | 781 | **$1.70 \times 10^{-3}$** | 993 | $1.62 \times 10^{-2}$ | 2,556 | **$4.00 \times 10^{-3}$** | 2,029 |
| I.12.2 | [4,3,3,1] | $2.83 \times 10^{-3}$ | 223 | **$1.84 \times 10^{-3}$** | 480 | $1.19 \times 10^{-2}$ | 751 | **$2.03 \times 10^{-3}$** | 672 |
| I.12.4 | [3,3,2,1] | $1.87 \times 10^{-3}$ | 70 | **$1.04 \times 10^{-3}$** | 159 | $4.55 \times 10^{-3}$ | 550 | **$6.13 \times 10^{-4}$** | 665 |
| I.12.5 | [2,2,1] | $1.32 \times 10^{-3}$ | 57 | **$1.16 \times 10^{-4}$** | **30** | $2.97 \times 10^{-2}$ | 201 | **$2.24 \times 10^{-2}$** | 330 |
| I.10.7 | [3,4,4,1,1] | $6.96 \times 10^{-3}$ | 307 | **$5.98 \times 10^{-3}$** | **115** | $9.77 \times 10^{-3}$ | **1,132** | **$4.00 \times 10^{-3}$** | 1,329 |
| I.12.11 | [5,2,2,1] | $8.73 \times 10^{-2}$ | 149 | **$8.55 \times 10^{-2}$** | **85** | $2.58 \times 10^{-1}$ | **565** | **$1.35 \times 10^{-1}$** | 1,312 |
| I.13.4 | [4,2,1,1] | $4.22 \times 10^{-3}$ | 103 | **$3.52 \times 10^{-3}$** | **55** | $1.84 \times 10^{-1}$ | **400** | **$1.21 \times 10^{-1}$** | 1,955 |
| I.13.12 | [5,4,4,1] | $3.41 \times 10^{-3}$ | 369 | **$2.83 \times 10^{-3}$** | **100** | $1.77 \times 10^{-2}$ | 1,489 | **$3.09 \times 10^{-3}$** | **1,336** |
| I.14.3 | [3,2,1] | $4.53 \times 10^{-3}$ | 75 | **$4.01 \times 10^{-3}$** | **40** | $4.66 \times 10^{-3}$ | **307** | $5.73 \times 10^{-3}$ | 332 |
| I.14.4 | [2,2,1,1] | $1.27 \times 10^{-3}$ | 67 | **$2.08 \times 10^{-4}$** | **35** | $1.16 \times 10^{-2}$ | 277 | $1.69 \times 10^{-2}$ | 319 |
| I.15.3x | [4,3,3,1] | $1.57 \times 10^{-2}$ | 223 | **$5.38 \times 10^{-3}$** | **80** | $2.59 \times 10^{-1}$ | 967 | **$2.33 \times 10^{-1}$** | **181** |
| I.15.10 | [3,3,2,1] | $8.45 \times 10^{-3}$ | 159 | **$7.79 \times 10^{-3}$** | **70** | $7.93 \times 10^{-3}$ | 703 | $8.38 \times 10^{-3}$ | 1,313 |
| I.18.4 | [4,4,3,1] | $2.21 \times 10^{-3}$ | 287 | **$1.55 \times 10^{-3}$** | **90** | $4.63 \times 10^{-3}$ | 1,310 | $8.19 \times 10^{-3}$ | **189** |

effectiveness of MetaWavKAN (Table 7), MetaFastKAN (Table 6), and MetaConvKAN (Table 3), indicating the universal applicability of MetaKANs framework.

**Memory efficiency analysis.** For a general KANs model, the additional parameter count arises from the learnable univariate functions $\phi(x; \mathbf{w})$, with the total parameters for a network structure $[n_0, n_1, \ldots, n_L]$ given by: $\sum_{l=0}^{L-1}(n_l \times n_{l+1}) \times \dim(\mathbf{w})$, where $\dim(\mathbf{w}) = G + k + 1$ for standard KANs. With meta-learner, the parameter count for KAN variants is reduced by approximately $1/\dim(\mathbf{w})$. For example, the dimensionality $\dim(\mathbf{w})$ is $c$ for FastKAN and 3 for WavKAN. The meta-learner generates activation parameters, reducing the total parameter count to approximately as in Eq.(3.2.3) which is comparable to that of an MLP. This makes MetaKANs and its variants significantly more memory efficient while retaining their flexibility and performance. Detailed comparisons are provided in Table 1.

### 3.4. More Discussion on the Memory Efficiency

It's important to understand the conditions that MetaKANs improve memory efficiency of KANs. KAN's parameters scale as $\sum_{l=0}^{L-1}(n_l \times n_{l+1}) \times (G + k + 1)$, reflecting total number of activations multiplied by the number of parameters per spline. MetaKANs generate these spline parameters with one prompt per activation ($\sum_{l=0}^{L-1}(n_l \times n_{l+1})$ total) but adds a fixed meta-learner cost ($\approx C(d_{hidden} + 1)(G + k + 1)$). Parameter reduction occurs when this fixed cost is less than the total spline parameters saved. This advantage is more pronounced for larger and deeper networks. By simple computation, we roughly require that $d_{hidden} \gtrsim \frac{G+k}{G+k+1} \sum_{l=0}^{L-1}(n_l \times n_{l+1})$ to ensure the memory efficiency of MetaKANs (see Figure 4). Consequently, for very small KANs in Table 2 with few total activations, the meta-learner's fixed cost can cause MetaKANs to have

slightly more parameters than KAN, despite MetaKANs often maintain competitive performance.

## 4. Experiments

We conduct extensive experiments to evaluate the performance of our method, including function fitting tasks in both low-dimensional (Sec.4.1) and high-dimensional scenarios (Sec.C.1), solving partial differential equations (Sec.C.3), and image classification tasks using KANs with both fully-connected (Sec.C.2) and convolutional architectures (Sec.4.2). Additionally, we provide the analysis of memory usage in Sec.4.3, and perform ablation study in Sec.C.4 to demonstrate the scalability of our approach with respect to prompt dimensions and layers of KANs.

### 4.1. Function Fitting Task

#### 4.1.1. EXPERIMENTAL SETUP

To demonstrate that MetaKANs reduce the overall number of trainable parameters without compromising the interpretability and function-fitting capabilities of the original KANs model, we conducted a function-fitting task on the Feynman dataset (Udrescu & Tegmark, 2020) as done in the original KAN paper. The comparison focused on the Mean Squared Error (MSE) performance.

In the experiments, we employed the LBFGS optimizer with an initial learning rate set to 1, consistent with the original paper. Additionally, to better balance fitting performance and model complexity, the hidden layer nodes of the meta-learner of MetaKANs were set to 32, 64. The number of grid points in set to $G = 5, 20$. The architecture of the KANs, including its depth and width, was designed based on the complexity and characteristics of each target function, following the principles outlined in (Liu et al., 2024).

*Table 3.* Performance Comparison on MNIST, CIFAR10, and CIFAR100 (4 and 8 Layers).* on 8-layer MetaKANConv and MetaFastKAN-Conv means we set the dimension of learnable prompts to 2 and 4 respectively and using the multiple meta-learners.

| MODEL | MNIST | | CIFAR-10 | | CIFAR-100 | |
|---|---|---|---|---|---|---|
| | # PARAM | ACC. | # PARAM | ACC. | # PARAM | ACC. |
| **4 LAYERS** | | | | | | |
| KANCONV | 3,489,774 | **98.43**± 0.46 | 3,494,958 | 41.92± 1.87 | 3,518,088 | 7.69± 0.34 |
| METAKANCONV | 391,255 | 96.03± 1.15 | **392,887** | **45.97**± 6.22 | 416,017 | **9.71**± 1.02 |
| FASTKANCONV | 3,489,292 | **99.36**± 0.02 | 3,494,480 | **68.12**± 2.85 | 3,517,610 | **34.64**± 5.02 |
| METAFASTKANCONV | 391,829 | 98.54± 0.51 | 392,409 | 66.69± 1.61 | 415,539 | 32.11± 2.98 |
| KALNCONV | 1,940,330 | 85.64± 4.28 | 1,943,210 | 32.98± 4.25 | 1,966,340 | 10.48± 0.61 |
| METAKALNCONV | **391,119** | **97.64**± 1.82 | **391,919** | **42.46**± 11.00 | 416,393 | **11.78**± 3.08 |
| KAGNCONV | 1,940,346 | 99.15± 0.11 | 1,943,226 | 72.08± 0.69 | 1,966,356 | **36.61**± 0.49 |
| METAKAGNCONV | 391,359 | **99.21**± 0.11 | 392,383 | **73.49**± 0.95 | 415,513 | 35.54± 2.49 |
| **8 LAYERS** | | | | | | |
| KANCONV | 40,694,018 | **99.50**± 0.08 | 40,696,610 | 67.24± 3.41 | 40,742,780 | 35.97± 2.84 |
| METAKANCONV | 4,532,170 | 99.46± 0.08 | 9,053,531 | **72.20**± 2.97 | 9,099,371 | **44.17**\* ± 3.19 |
| FASTKANCONV | 40,693,052 | **99.71**± 0.02 | 40,695,648 | **79.84**± 0.46 | 40,741,818 | 50.80± 0.17 |
| METAFASTKANCONV | 4,539,652 | 99.42 ± 0.08 | 9,051,033 | 78.73± 1.59 | 18,142,821 | **52.02**\* ± 1.07 |
| KALNCONV | 22,611,642 | 69.02± 10.54 | 22,613,082 | 29.14± 1.67 | 22,659,252 | 9.10± 0.41 |
| METAKALNCONV | 4,529,727 | **99.53**± 0.05 | 4,530,015 | **70.23**± 2.43 | 4,576,185 | **21.94**± 5.69 |
| KAGNCONV | 22,611,674 | **99.58**± 0.06 | 22,613,114 | **84.69**± 0.74 | 22,659,284 | 56.41± 0.32 |
| METAKAGNCONV | 4,543,682 | 99.46± 0.20 | 4,530,495 | 84.25± 0.36 | 4,576,665 | **57.76**± 0.50 |

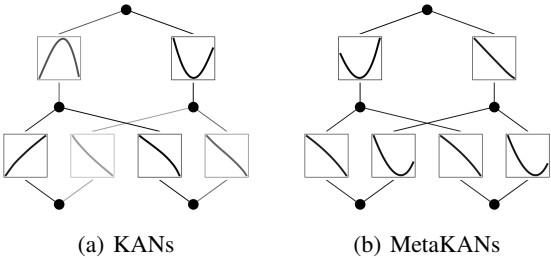

(a) KANs                    (b) MetaKANs

*Figure 3.* Function fitting on I.12.5 ($f(x_1, x_2) = x_1 x_2$). KANs learn the formula by $f(x_1, x_2) = -(x_1 - x_2)^2 + (-x_1 - x_2)^2 = 2x_1 x_2$, while MetaKANs learn the formula by $f(x_1, x_2) = (-x_1 - x_2)^2 - (x_1^2 + x_2^2) = 2x_1 x_2$.

### 4.1.2. RESULTS

The comprehensive evaluation in Table 2 demonstrates MetaKANs' significant advantage in balancing parameter efficiency and approximation accuracy. Across 16 Feynman equations, MetaKANs consistently outperform the baseline KANs model under identical grid configurations while using substantially fewer parameters. At $G = 5$, MetaKAN achieves lower MSE in 11 out of 17 functions with trainable parameter reductions ranging from 21% to 47%. A notable example is the I.12.5 equation($f(x_1, x_2) = x_1 x_2$), where MetaKANs attains an order-of-magnitude improvement in MSE ($1.16 \times 10^{-4}$ vs $1.32 \times 10^{-3}$) while requiring only 30 parameters compared to 57 parameters of KANs (learned symbolic functions refer to Fig. 3). The results persist at higher grid resolutions, with MetaKANs maintaining superior accuracy in 13/16 cases at $G = 20$ despite using 18% fewer parameters on average. To further illustrate the insight behind our meta-learner design, we also visualize the parameters' inner dot similiarity matrix of each activation function for KANs and MetaKANs (with

structure $[4, 5, 5, 1]$) respectively on fitting the function $f(\mathbf{x}) = \exp\left(\frac{1}{2} \left(\sin\left(\pi \left(x_1^2 + x_2^2\right)\right) + \sin\left(\pi \left(x_3^2 + x_4^2\right)\right)\right)\right)$ in Figure 9. From the figure, we observe that the functional class learned by MetaKANs is significantly more compact compared to that learned by KANs. This indicates that fitting the target function requires a much smaller function space, whereas KANs learn a larger and redundant function space, leading to memory inefficiency. More detailed analysis please refers to Section B.

### 4.2. Convolutional Architecture Experiments

#### 4.2.1. EXPERIMENTAL SETUP

Extending the framework from (Drokin, 2024), we implement four core modules: KANConv, FastKANConv, KAG-NConv, and KALNConv. The baseline architecture contains two configurations: 4-layer (shallow) and 8-layer (deep) networks. Channel progression follows [32, 64, 128, 512] for 4-layer and [2, 64, 128, 512, 1024, 1024, 1024, 1024] for 8-layer architectures, with 3×3 kernels, stride 1, and padding 1. And for 8-layer MetaKANConv and MetaFastKANConv, we set the dimension of learnable prompts to 2 and 4 respectively and using the multiple meta-learners. The details of clustering meta-learner is stated in Sec. C.4.

The optimization strategy employs three AdamW optimizers: learnable prompts ($\eta = 10^{-4}$), meta-learner ($\eta = 10^{-3}$), and main network ($\eta = 10^{-4}$). Training details include random horizontal flipping and cropping for CIFAR datasets, exponential decay learning rate scheduling, and Dropout ($p = 0.2$) regularization.

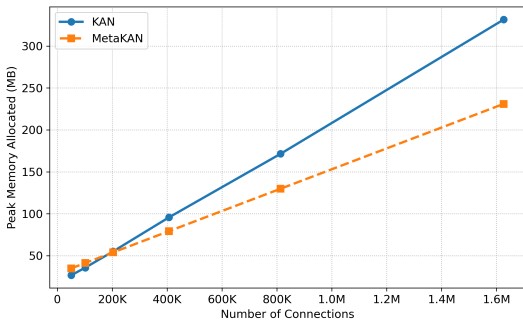

(a) Memory usage vs. number of connections

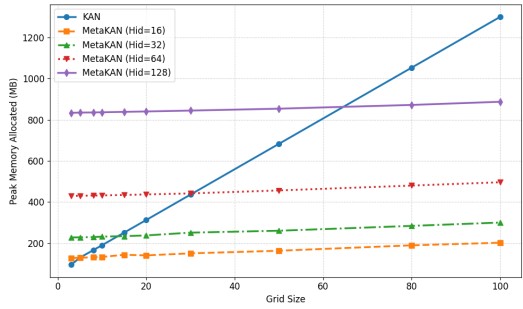

(b) Memory usage vs. grid sizes

*Figure 4.* Memory efficiency of MetaKANs compared to KANs. Peak memory allocation (MB) vs. network connections for grid size $G = 5$ (Top) and vs. grid sizes for hidden dim $d_{\text{hidden}} = 64$ (Bottom). MetaKANs show substantial memory savings.

### 4.2.2. EXPERIMENTAL RESULTS

The convolutional architecture comparisons (shown in Table 3) reveal significant parameter efficiency gains while maintaining competitive accuracy across datasets. For 4-layer models, MetaKANConv achieves 45.86% accuracy on CIFAR-10 with 89% parameter reduction (392,887 vs 3,494,958 parameters), outperforming the baseline by 6.03 percentage points. MetaFastKANConv demonstrates particularly strong results on CIFAR-100, attaining 34.81% accuracy (vs 28.84% baseline) using only 12% of original parameters (415,539 vs 3,517,610). The 8-layer configurations show enhanced performance scaling, with MetaFastKANConv reaching 81.53% on CIFAR-10 while reducing parameters by 78% (9,051,033 vs 40,695,648).

### 4.3. Benefits to Memory Usage

A primary goal of MetaKANs is reducing KAN's high training costs, particularly peak GPU memory consumption. Figure 4 shows the peak memory usage of MetaKANs and KANs under different network sizes and grid settings. This demonstrates that MetaKANs are significantly more memory-efficient, especially as the size of network grows larger or grid size increases. MetaKANs scale better and keep memory consumption lower compared to KANs.

Its effectiveness in saving memory is particularly pro-

nounced for KANs with large grid sizes, such as $G = 80$ and above. In these high-resolution scenarios, MetaKANs provide substantial memory savings compared to standard KANs, even when the MetaKANs use larger hidden dimensions for the meta-learner. This makes it well-suited for developing scalable, high-resolution KANs model. For KANs with medium grid sizes, roughly $G = 20$ to $G = 50$, MetaKANs can also effectively reduce memory, especially if the meta-learner is configured with a small to moderate hidden dimension, like 16 or 32. Similarly, in settings with small grid sizes, such as $G = 5$ to $G = 15$, memory savings are achievable if a meta-learner with a very small hidden dimension (e.g., 16) is employed. Therefore, MetaKANs offer a valuable strategy for making KANs more memory-efficient, particularly for high grid resolution. For a practical suggestion, the hidden dimension of the meta-learner should be chosen in consideration of the KAN's grid size.

### 4.4. Physical Meaning of Learned Prompt Embedding

Figure 10 visualizes the learned task prompt $\mathcal{Z}$ defined in Eq.(11) to understand whether $\mathcal{Z}$ can distinguish activation functions with different task properties for basis functions. The heatmap (right) shows the similarity of different task prompts $z_\alpha^{(l)}, \alpha \in [n_l] \times [n_{l+1}]$ (left). We can see that learned task prompts for similar activation functions are more similar (darker regions in the heatmap) than that for dissimilar activation functions. This suggests that learnable task prompt could identify which activation function it is, validating the rationality for the design of the meta-learner. This further supports that proposed MetaKANs are capable of learning proper weighting learning rules for KANs, and thus improving memory efficiency.

## 5. Conclusion

In this paper, we introduced MetaKANs, a memory-efficient framework for training KANs through meta-learning. By leveraging a smaller meta-learner to generate weights for KANs, MetaKANs significantly reduce the number of trainable parameters while maintaining or even improving performance. Our experiments across various benchmark tasks, including symbolic regression, partial differential equation solving, and image classification, demonstrate that MetaKANs achieve comparable or superior performance with only a fraction of the parameters required by traditional KANs. Furthermore, MetaKANs are model-agnostic, making them applicable to a wide range of KAN variants, such as FastKAN, ConvKAN, and WavKAN. In conclusion, MetaKANs provide a promising direction for improving the training efficiency of KANs, enabling their application to larger datasets and more complex tasks. Besides, how to transfer the weights prediction rules for helping improve memory efficiency of novel KANs training tasks is a potentially valuable future research direction.

# Acknowledgements

Authors acknowledge the constructive feedback of reviewers and the work of ICML'25 program and area chairs. This work was supported by the National Key Research and Development Program of China (2022YFA1004100) and in part by the National Natural Science Foundation of China (62476214, 12326606, U24A20324 and 32430017), Tianyuan Fund for Mathematics of the National Natural Science Foundation of China (Grant No. 12426105) and the Major Key Project of PCL under Grant PCL2024A06.

# Impact Statement

In this work, we explore to understand the working mechanism of KANs and reveal the potential inefficiency of weights learning for KANs. Based on this observation, we improve memory efficiency of training KANs by meta-learning techniques, and achieve almostly the same trainable parameters with MLP network. We believe such attempt will inspire future research toward bridging the gap of training cost between KANs and MLP network. It is meaningful to make KANs easier to learn with limited resources while keeping their interpretability and better performance. However, we note that both of our algorithm and KANs may perform worse on image classification benchmarks than existing baselines. Some explanations for this phenomenon and possible strategis for addressing the issue are worthwhile to explore.

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

# A. Extension to Other KAN Variants

## A.1. WavKAN

In WavKAN, (Bozorgasl & Chen, 2024) propose replacing the activation functions in KAN with mother wavelet functions. This modification aims to enhance the model's multi-scale resolution capability, allowing it to effectively capture features across different scales. Additionally, the use of wavelet functions reduces the overall parameter count compared to traditional KAN, as the wavelet basis functions inherently introduce sparsity and compactness into the representation.

Specifically, taking the Mexican hats wavelet function as an example, it follows as:

$$\psi(t;\sigma) = \frac{2}{\pi^{1/4}\sqrt{3\sigma}} \left( \frac{t^2}{\sigma^2} - 1 \right) \exp\left( \frac{-t^2}{2\sigma^2} \right). \quad (16)$$

where $\sigma$ shows the adjustable standard deviation of Gaussian. As for each activation function

$$\phi(t;\mathbf{w}) = w\psi(t - \mu;\sigma) \quad (17)$$

where $w$ is a learnable scaling factor controlling the amplitude of the wavelet function, and $\mu$ represents the translation parameter that adjusts the center of the wavelet. The parameters are compactly represented as $\mathbf{w} = [w, \mu, \sigma]^T$. Thus, each activation function in WavKAN is parameterized by just three learnable parameters $\mathbf{w} \in \mathbb{R}^3$. For the WavKAN, it is easily to replace the activation function in Eq. (2) with the $\phi(t;\mathbf{w}) = w\psi(t - \mu;\sigma)$.

As a result, the parameter count for WavKAN is significantly reduced compared to KAN. Specifically, for structure $[n_0, n_1, ..., n_L]$, WavKAN requires $\mathcal{O}\left( 3 \times \sum_{l=0}^{L-1}(n_l \times n_{l+1}) \right)$ parameters, which is approximately $3\times$ the parameter count of a standard MLP with the same structure.

Although WavKAN reduces the parameter count compared to KAN, it still introduces additional parameters relative to MLPs, especially when the network structure becomes more complex. To address this, we propose meta-learner, referred to as MetaWavKAN, to generate the parameters of WavKAN, further reducing the overall parameter count. Specifically, the meta-learner $M_\theta(z) : \mathbb{R} \to \mathbb{R}^3$ generates the three learnable parameters $\mathbf{w}(z, \theta) = [w, \mu, \sigma]^T = M(z; \theta)$ for each activation function with one learnable scalar $z$. The meta-learner $M_\theta(z)$ is the same two-layer MLP used in MetaKANs. Consequently, MetaWavKAN maintains the same total parameter count as MetaKAN, as described in Sec. (3.2.3).

## A.2. FastKAN

FastKAN (Li, 2024) proposes an efficient approximation method for B-splines of order $k$ with $G$ intervals by employing a Radial Basis Function (RBF) network. An RBF network approximates a target function using a weighted sum of radial basis functions centered at specific points. The general form of an RBF network can be expressed as:

$$f(x; \mathbf{w}) = \sum_{i=1}^{G+k} w_i \phi\left( \|x - c_i\| \right) = \mathbf{w}^T \mathbf{B}(x) \quad (18)$$

where $\mathbf{w} = [w_1, ..., w_{G+k}]^\top$, and $\mathbf{B}(x) = [\phi(\|x - c_1\|), ..., \phi(\|x - c_{G+k}\|)]^\top$.

Gaussian RBF is specifically chosen as the basis function due to its smoothness and universal approximation capabilities. The Gaussian RBF is defined as:

$$\phi(r) = \exp\left( -\frac{r^2}{2h^2} \right),$$

where $r$ is the radial distance, and $h$ is a parameter that controls the width or spread of the function.

By selecting $G+k$ centers for the Gaussian RBFs, FastKAN efficiently approximates a series of B-spline basis functions, offering both computational efficiency and strong function approximation capabilities. For the consistency of the formulation, B-spline basis function in Eq. (2) could be substituted with Eq. (18).

Although FastKAN improves computation efficiency via RBF approximation, it still requires $(G + k)\times$ parameters count compared with MLP. Thus similar to MetaKAN and MetaWavKAN, we propose using meta-learner, named MetaFastKAN, to generate parameters $\mathbf{w}$ in the Eq. 18.

## A.3. ConvKAN

Kolmogorov-Arnold convolutions, introduced by (Bodner et al., 2024) and extended by (Drokin, 2024), which replace traditional convolutional activation functions with Kolmogorov-Arnold-based transformations, such as spline (2), RBF (18), or wavelet (17).

Unlike conventional convolution kernels, Kolmogorov-Arnold (KA) kernels consist of a set of univariate nonlinear learnable activation functions. Given an image input $X \in \mathbb{R}^{c \times h \times w}$ and a KA kernel $K \in \mathbb{R}^{N \times M}$, the KA convolution operation can be expressed as:

$$y_{ij} = \sum_{d=1}^{c} \sum_{a=0}^{k-1} \sum_{b=0}^{k-1} \phi_{a,b,d}(x_{d,i+a,j+b}; \mathbf{w}_{a,b,d}), \quad (19)$$

where $\phi$ represents a univariate non-linear learnable function with trainable parameters $\mathbf{w}$. This function can take

forms such as B-splines (KANConv), wavelet functions (WavKANConv), or Gaussian RBFs (FastKANConv), depending on the chosen architecture. However, similar to conventional convolutions, the inclusion of additional parameters $\mathbf{w}$ leads to inefficiency in parameter storage.

To address this inefficiency, we propose introducing a meta-learner to in-context learn the function class and generate the parameters. Specifically, the learnable parameters consist of prompts $\mathcal{Z}$ for each layer, as well as the kernel size and the number of input and output channels.

For a ConvKAN with $L$ layers structured as $[n_0, n_1, \ldots, n_L]$, we introduce a set of prompts $\mathcal{Z}$ for each layer, denoted as:

$$\mathcal{Z} = \bigcup_{l=1}^{L-1} \left\{ z_\alpha^{(l)} \mid \alpha = (i,j,a,b) \in \atop [n_l] \times [n_{l+1}] \times [K_l]^2 \right\} \tag{20}$$

where $K_l$ represents the kernel size of the $l$-th layer, and $a$ and $b$ are the kernel indices. These prompts $\mathbf{z}^{(l)}$ are used by the meta-learner to generate the weights $\mathbf{w}$ for each convolutional layer. The number of learnable parameters depends on the kernel size $K_l$ and the input-output channel dimensions.

Specifically, for each layer, the weight generation process is formulated as:

$$\mathcal{W}(\mathcal{Z}, \theta) = \bigcup_{l=0}^{L-1} \left\{ \mathbf{w}_\alpha^{(l)} = M_\theta(z_\alpha^{(l)}) \mid \alpha \in \mathcal{I}_l \right\}, \tag{21}$$

where $\mathcal{I}_l = [n_l] \times [n_{l+1}] \times [K_l]^2$, $\quad l \in [L] - 1$. The hypernetwork $M_\theta$ maps the embedding scalars $z_{i,j,a,b}^{(l)}$ to the corresponding convolution kernel parameters $\mathbf{w}_{i,j,a,b}^{(l)}$.

As discussed in Section 3.3, the parameter count of Meta-ConvKAN can be reduced by a factor of $1/\texttt{dim}(\mathbf{w})$ compared to KANConv, resulting in a parameter efficiency approximately equivalent to that of standard convolutional networks. If we set the dimension of the $z$ from 1 to $\texttt{dim}(z)$, the parameter count can be reduced by a factor of $\texttt{dim}(z)/\texttt{dim}(\mathbf{w})$ compared to KANConv.

## B. Illustration of the Learned Function Class

We train KANs and MetaKANs (with structure $[4, 5, 5, 1]$) respectively on fitting the function $f(\mathbf{x}) = \exp\left(\frac{1}{2}\left(\sin\left(\pi\left(x_1^2 + x_2^2\right)\right) + \sin\left(\pi\left(x_3^2 + x_4^2\right)\right)\right)\right)$ and the parameter inner dot similarity matrices are represented in Figure 9. From the figure, we observe that the function class learned by MetaKANs is significantly more compact compared to that learned by KANs. This indicates that MetaKANs require a smaller set of function class to fit the target function, whereas KANs learn a

relatively larger and redundant function class, leading to memory inefficiency.

For KANs, it achieves the MSE error $4.58e - 02$, while MetaKANs achieve MSE error $3.54e - 02$. After applying node pruning and symbolic regression as introduced in (Liu et al., 2024), we can formulate the learned symbolic function. For KANs, the result is

$$\begin{aligned}
\mathrm{KAN}(\mathbf{x}; \mathcal{W}) &= -0.0311\big(-x_3 + 0.2892x_4 + 0.6304\big)^2 \\
&- 0.0118\big(x_3 + 0.4825x_4 - 0.3973\big)^2 \\
&+ 0.6505\Big(0.0509x_3 - 0.0147x_4 \\
&\quad - 0.3847\sin(0.996x_3 - 7.7876) \\
&\quad - 0.1353\sin(2.0286x_4 - 1.5882) \\
&\quad - \sin\big(2.281\sin(1.4001x_1 + 1.576) \\
&\qquad - 2.2739\sin(1.4004x_2 - 1.572) + 7.5942\big) \\
&\quad + 0.6822\sin(-0.0424x_1 + 0.04x_2 \\
&\qquad - 0.1144x_3 + 0.0639x_4 + 8.2417) - 0.8938\Big)^2 \\
&+ 0.3489\sin(1.4001x_1 + 1.576) \\
&- 0.3478\sin(1.4004x_2 - 1.572) \\
&- 0.6128\sin(0.996x_3 - 7.7876) \\
&- 0.2155\sin(2.0286x_4 - 1.5882) \\
&+ 0.322\sin(-0.0424x_1 + 0.04x_2 \\
&\qquad - 0.1144x_3 + 0.0639x_4 + 7.8417) - 0.4717,
\end{aligned}$$

while the result of MetaKAN is

$$\begin{aligned}
\mathrm{MetaKAN}(\mathbf{x}; \mathcal{W}) &= 1.019\sin\bigg(0.5264\sin \\
&\big(3.1711(0.0002 - x_4)^2 \\
&\quad + 3.1753(0.0004 - x_3)^2 - 12.5921\big) \\
&+ 0.525\sin\big(3.1597(0.0004 - x_2)^2 \\
&\quad + 3.1578(0.0005 - x_1)^2 - 12.5819\big) \\
&- 9.8227\bigg) - 1.8769.
\end{aligned}$$

The results demonstrate that MetaKAN not only achieves better accuracy but also learns a more compact and interpretable representation of the target function. Specifically, MetaKAN's formula is significantly simpler, with fewer terms and a clearer structure. This highlights MetaKAN's advantage in learning concise and generalizable representations, which is crucial for both interpretability and memory efficiency.

*Table 4.* Parameter counts and MSE for KANs and MetaKANs at various dimensions for three different functions.

| FUNCTION | $f_1(x) = \exp\left(\frac{1}{n}\sum \sin^2\left(\frac{\pi x}{2}\right)\right)$ | | | | $f_2(x) = \sum x^2 + x^3$ | | | | $f_3(x) = \exp\left(-\frac{1}{n}\sum x^2\right)$ | | | |
|---|---|---|---|---|---|---|---|---|---|---|---|---|
| DIMENSION | KAN | | METAKAN | | KAN | | METAKAN | | KAN | | METAKAN | |
| STRUCTURE [N,1,1] | # PARAM | MSE | # PARAM | MSE | # PARAM | MSE | # PARAM | MSE | # PARAM | MSE | # PARAM | MSE |
| 10 | 101 | 5.92E-4 | 97 | **5.85E-4** | 191 | 3.11E-3 | 119 | **7.06E-4** | 101 | **1.47E-4** | 97 | 3.39E-4 |
| 20 | 191 | **4.63E-4** | 108 | 4.97E-4 | 371 | 2.63E-3 | 141 | **2.26E-4** | 191 | **1.11E-5** | 108 | 3.42E-5 |
| 50 | 461 | **5.52E-4** | 130 | 1.40E-3 | 911 | NA | 185 | **1.15E-3** | 461 | 3.46E-4 | 174 | **3.19E-4** |
| 100 | 911 | 7.00E-3 | 229 | **1.45E-3** | 1811 | 1.74 | 361 | **1.16E-2** | 911 | **5.12E-4** | 229 | 6.47E-4 |
| 500 | 4,511 | 2.46E-1 | 339 | **4.62E-3** | 9011 | 1.83E+2 | 889 | **5.52E-2** | 4,511 | 3.69E-2 | 669 | **8.83E-3** |
| 1000 | 9,011 | 4.14E-1 | 713 | **1.48E-2** | 18,011 | 1.68E+2 | 1329 | **1.43E-1** | 9,011 | 9.70E-2 | 2,759 | **3.94E-3** |

## C. Extended Experiments

### C.1. High Dimensional Function

#### C.1.1. EXPERIMENT SETUP

In this experiment, we evaluated the performance of the KANs and MetaKANs models on three different high-dimensional functions. The target functions were defined as follows:

$$f_1(x) = \exp\left(\frac{1}{n}\sum \sin^2\left(\frac{\pi x}{2}\right)\right)$$

$$f_2(x) = \sum x^2 + x^3$$

$$f_3(x) = \exp\left(-\frac{1}{n}\sum x^2\right)$$

For each function, the models were evaluated at six different dimensions: $n = 10, 20, 50, 100, 500, 1000$. Experiments span dimensions $n = 10$ to $1000$ using a minimal $[n, 1, 1]$ architecture. For MetaKAN, the hypernetwork's hidden layer width was systematically explored from 2 to 64 neurons through grid search, with optimal widths selected as $\{8, 16, 32, 32, 64, 64\}$ corresponding to dimensions $\{10, 20, 50, 100, 500, 1000\}$ respectively. This controlled expansion ensures the hypernetwork's capacity grows proportionally with input dimension while maintaining parameter efficiency.

The training was conducted using the LBFGS optimizer, and the results were measured in terms of parameter count and MSE. The tables display the parameter counts and corresponding MSE values for both KAN and MetaKAN models across all dimensions for each function.

#### C.1.2. RESULTS

Table 4 demonstrates MetaKAN's superiority in high-dimensional function approximation across three key aspects. Across all function types and dimensions, MetaKAN demonstrates superior accuracy with significantly fewer parameters. For the cubic function $f_2$ at 500 dimensions, MetaKAN achieve a 330× lower MSE ($5.52 \times 10^{-2}$ vs $1.83 \times 10^2$) while using only 10% of KAN's parameters (889

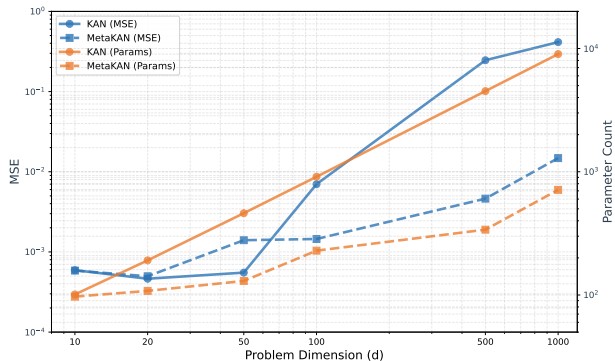

*Figure 5.* Function $f_1$: Accuracy and trainable parameter count under different dimension

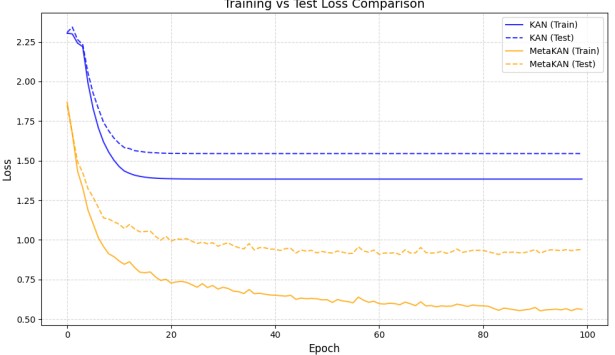

*Figure 6.* Training and test Loss comparison for KANs and MetaKANs on SVHN dataset ($G = 5$)

vs 9,011). This efficiency advantage amplifies with dimensionality - for the radial function $f_3$, MetaKAN's parameter count grows 28× from 10D to 1000D (97 to 2,759) versus KAN's 89× increase (101 to 9,011), enabling MetaKAN to limit error growth to 26.8× compared to KAN's 658× MSE escalation. Critical limitations emerge for KAN: failure to converge for $f_2$ at 50 dimensions, and error amplification to $1.68 \times 10^2$ MSE at 1000D with 18,011 parameters, versus MetaKAN's stable $1.43 \times 10^{-1}$ MSE using 1,329 parameters. These patterns are visually corroborated in Figure 5, demonstrating MetaKANs consistently perform better on different dimension settings while require less parameter count than KANs.

*Table 5.* Classification accuracies and parameter counts for KAN and MetaKAN models across different settings.

| | | G=3 | | | | G=5 | | | | G=10 | | | |
| | | KAN | | METAKAN | | KAN | | METAKAN | | KAN | | METAKAN | |
| DATASET | HIDDEN | # PARAM | ACC. | # PARAM | ACC. | # PARAM | ACC. | # PARAM | ACC. | # PARAM | ACC. | # PARAM | ACC. |
| --- | --- | --- | --- | --- | --- | --- | --- | --- | --- | --- | --- | --- | --- |
| SVHN | 32 | 529,130 | 51.16 | **77,376** | **78.34** | 680,298 | 52.35 | **78,144** | **75.35** | 831,466 | 59.67 | 79,424 | **63.46** |
| FMNIST | 32 | 177,898 | 84.59 | **27,456** | **87.28** | 228,714 | 85.59 | **27,968** | **87.73** | 355,754 | 85.65 | 29,248 | **86.7** |
| KMNIST | 32 | 177,898 | 77.64 | **27,456** | **83.93** | 228,714 | 78.28 | **27,968** | **82.82** | 355,754 | 73.52 | 29,248 | **73.22** |
| MNIST | 32 | 177,898 | 94.11 | **27,456** | **96.3** | 228,714 | 93.60 | **27,968** | **96.69** | 355,754 | 92.76 | 29,248 | **95.43** |
| CIFAR-10 | 32 | 529,130 | 39.36 | **77,632** | **47.98** | 680,298 | 36.75 | **78,144** | **49.67** | 1,058,218 | 39.86 | 79,424 | **48.36** |
| CIFAR-100 | 32 | 549,380 | 4.45 | **80,512** | **21.09** | 706,308 | 8.75 | **81,024** | **20.85** | 1,098,628 | 5.66 | 82,304 | **19.35** |
| SVHN | 32,32 | 536,330 | 60.31 | **78,656** | **78.10** | 689,546 | 62.54 | **79,168** | **77.01** | 1,072,586 | 58.71 | 80,448 | **59.51** |
| FMNIST | 32,32 | 185,098 | 85.91 | **28,480** | **87.6** | 237,962 | 86.12 | **28,992** | **86.16** | 370,122 | **86.73** | 30,272 | 85.81s |
| KMNIST | 32,32 | 185,098 | 80.46 | **28,480** | **84.26** | 237,962 | 80.23 | **28,992** | **83.44** | 370,122 | **81.44** | 30,272 | 80.07 |
| MNIST | 32,32 | 185,098 | 93.80 | **28,480** | **95.91** | 237,962 | 91.22 | **28,992** | **94.43** | 370,122 | 93.32 | 30,272 | **94.87** |
| CIFAR-10 | 32,32 | 536,330 | 42.16 | **78,656** | **49.75** | 689,546 | 47.15 | **79,168** | **49.4** | 1,072,586 | 43.79 | 80,448 | **44.6** |
| CIFAR-100 | 32,32 | 556,580 | 2.95 | **81,536** | **20.49** | 715,556 | 8.8 | **82,048** | **14.42** | 1,112,996 | 6.48 | 83,328 | **18.23** |

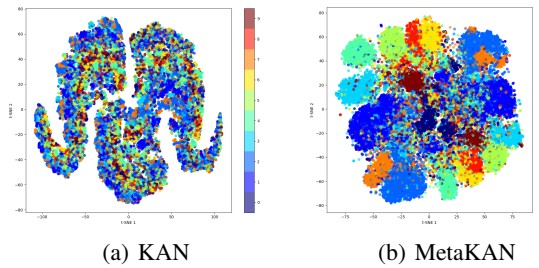

(a) KAN        (b) MetaKAN

*Figure 7.* T-SNE visualization on the SVHN test datset. The MetaKAN performs clearer classification boundaries

## C.2. Fully Connected Architecture Experiments

### C.2.1. EXPERIMENTAL SETUP

We systematically evaluated three base architectures (KAN, WavKAN, and FastKAN) along with their meta-learning variants. The network architecture employs single-hidden-layer (32 neurons) and double-hidden-layer (32×32 neurons) configurations. For activation function configuration: KAN uses B-spline basis functions with grid points $G \in \{3, 5, 10\}$ and polynomial order $k = 3$; WavKAN adopts three mother wavelet functions (Mexican Hat, Morlet, and Derivative of Gaussian); FastKAN sets the number of centers $c \in \{6, 8, 13\}$ to align with the B-spline basis count in KAN.

The meta-learner design involves hidden layer dimensions selected from $\{32, 64, 128\}$ through grid search. Output dimension adaptation follows specific rules: for KAN, $d_{out} = (G + k + 1)$; for WavKAN, $d_{out} = 3$ corresponding to the three wavelet bases; for fastkan, $d_{out} = c + 1$. To training MetaKANs , we use separate AdamW optimizers for the meta-learner $M$ ($\eta_1 \in \{10^{-4}, 10^{-3}\}$) and learnable prompts ($\eta_2 \in \{10^{-3}, 10^{-2}\}$). Training configuration employs cosine annealing for learning rate scheduling and batch size 128.

### C.2.2. RESULTS

The comparative results across different architectures reveal consistent patterns in parameter efficiency and performance gains. For standard KANs (Table 5), MetaKANs achieve $85 \sim 94\%$ parameter reduction while maintaining competitive accuracy, particularly evident in SVHN (78.34% vs 51.16% at $G = 3$) and FMNIST (87.28% vs 84.59%). We visualize the training procedure on the SVHN (Figure 6) and the t-SNE visualization (Figure 7).

The dual-hidden-layer configuration (32,32) demonstrates improved performance on complex datasets, with MetaKAN reaching 49.75% accuracy on CIFAR-10 compared to 42.16% for baseline KAN at $G = 3$.

The FastKAN experiments (Table 6) demonstrate MetaFastKAN's parameter efficiency with 84-92% reduction across configurations while maintaining or improving accuracy. For CIFAR-100 with dual hidden layers (32,32), MetaFastKAN achieves 23.17% accuracy at $c = 13$ using only 8.3% of baseline parameters (92,565 vs 1,117,867). The architecture shows particular strength in simpler datasets, attaining 96.31% accuracy on MNIST ($c = 6$) versus baseline's 94.99% with 15.7% parameter load. Notably, the dual-hidden-layer configuration achieves superior results on complex tasks, improving SVHN accuracy to 79.52% at $c = 13$ (vs 75.51% baseline) while maintaining 91.7% parameter reduction. Shallow architectures (single hidden layer) preserve competitive efficiency, reaching 88.29% accuracy on FMNIST ($c = 6$) with 16% of original parameters.

Wavelet-based architectures (Table 7) demonstrate mother function-dependent performance, with Mexican Hat wavelet achieving 73.62% accuracy on SVHN compared to 66.13% for DoG wavelet in MetaWavKAN. The dual-hidden-layer configuration improves generalization on CIFAR-10, where MetaWavKAN reaches 49.52% accuracy versus 46.34% in

*Table 6.* Classification accuracies and parameter counts for FastKAN and MetaFastKAN models across different settings.

| DATASET | HIDDEN | c=6 FASTKAN # PARAM | ACC. | c=6 METAFASTKAN # PARAM | ACC. | c=8 FASTKAN # PARAM | ACC. | c=8 METAFASTKAN # PARAM | ACC. | c=13 FASTKAN # PARAM | ACC. | c=13 METAFASTKAN # PARAM | ACC. |
|---|---|---|---|---|---|---|---|---|---|---|---|---|---|
| SVHN | 32 | 533,910 | 72.94 | 81,523 | **77.00** | 685,082 | 66.19 | 81,081 | **73.19** | 1,063,012 | 70.24 | 82,440 | **70.27** |
| FMNIST | 32 | 179,542 | 86.64 | 28,211 | **88.29** | 230,362 | 87.41 | 29,881 | **87.93** | 357,412 | **88.9** | 35,272 | 88.44 |
| KMNIST | 32 | 179,542 | 75.99 | 28,211 | **81.67** | 230,362 | 80.38 | 29,881 | **81.02** | 357,412 | 82.87 | 35,272 | **84.22** |
| MNIST | 32 | 179,542 | 92.98 | 28,211 | **96.29** | 230,362 | 95.16 | 27,769 | **95.31** | 357,412 | 94.82 | 29,128 | **95.72** |
| CIFAR-10 | 32 | 533,910 | 47.99 | 82,675 | **51.73** | 685,082 | 47.05 | 83,193 | **50.77** | 1,063,012 | 50.57 | 81,523 | **50.27** |
| CIFAR-100 | 32 | 554,160 | 22.28 | 87,859 | **23.19** | 711,092 | 22.36 | 88,889 | **23.5** | 1,103,422 | 21.56 | 87,368 | **22.26** |
| SVHN | 32,32 | 541,180 | 72.43 | 82,617 | **77.20** | 694,402 | 71.25 | 82,881 | **74.69** | 1,077,457 | 75.51 | 89,685 | **79.52** |
| FMNIST | 32,32 | 186,812 | 87.13 | 32,761 | **88.13** | 239,682 | **88.71** | 29,569 | 87.92 | 371,857 | **88.9** | 30,229 | 87.96 |
| KMNIST | 32,32 | 186,812 | 82.62 | 29,305 | **82.96** | 239,682 | **85.82** | 33,793 | 85.46 | 371,857 | **85.04** | 36,373 | 84.9 |
| MNIST | 32,32 | 186,812 | 94.99 | 29,305 | **96.31** | 239,682 | **96.37** | 29,569 | 95.99 | 371,857 | 95.47 | 30,229 | **96.37** |
| CIFAR-10 | 32,32 | 541,180 | 48.78 | 83,769 | **51.55** | 694,402 | 48.32 | 87,105 | **49.38** | 1,077,457 | 50.57 | 89,685 | **51.16** |
| CIFAR-100 | 32,32 | 561,430 | 18.5 | 86,649 | **22.6** | 720,412 | 17.87 | 89,985 | **22.47** | 1,117,867 | 21.08 | 92,565 | **23.17** |

*Table 7.* Classification accuracies and parameter counts for WavKAN and MetaWavKAN models across different settings.

| DATASET | HIDDEN | DOG WAVELET WAVKAN # PARAM | ACC. | DOG WAVELET METAWAVKAN # PARAM | ACC. | MORLET WAVELET WAVKAN # PARAM | ACC. | MORLET WAVELET METAWAVKAN # PARAM | ACC. | MEXICAN HAT WAVELET WAVKAN # PARAM | ACC. | MEXICAN HAT WAVELET METAWAVKAN # PARAM | ACC. |
|---|---|---|---|---|---|---|---|---|---|---|---|---|---|
| SVHN | 32 | 226,836 | 45.65 | 76,951 | **66.13** | 226,836 | 19.08 | 76,311 | **25.69** | 226,836 | 67.05 | 76,951 | **73.62** |
| FMNIST | 32 | 76,308 | 83.86 | 26,775 | **87.66** | 76,308 | 83.11 | 26,135 | **82.1** | 76,308 | 87.13 | 25,815 | **87.23** |
| KMNIST | 32 | 76,308 | 77.61 | 26,775 | **78.76** | 76,308 | 77.08 | 25,815 | **79.98** | 76,308 | **83.37** | 25,731 | 82.17 |
| MNIST | 32 | 76,308 | 94.58 | 26,775 | **95.3** | 76,308 | 93.14 | 25,815 | **96.17** | 76,308 | **96.78** | 30,531 | 95.9 |
| CIFAR-10 | 32 | 226,836 | 33.33 | 76,951 | **46.34** | 226,836 | 13.48 | 76,311 | **23.11** | 226,836 | 47.17 | 75,991 | **48.09** |
| CIFAR-100 | 32 | 235,656 | 11.11 | 80,011 | **20.96** | 235,656 | 1.5 | 79,371 | **5.71** | 235,656 | 20.51 | 83,587 | **21.27** |
| SVHN | 32,32 | 306,580 | 46.27 | 78,039 | **69.42** | 229,972 | 19.47 | 77,079 | **55.20** | 229,972 | 60.83 | 81,731 | **76.55** |
| FMNIST | 32,32 | 79,444 | 83.72 | 27,863 | **84.73** | 79,444 | 76.34 | 27,223 | **80.02** | 79,444 | 86.52 | 27,863 | **86.81** |
| KMNIST | 32,32 | 79,444 | 79.87 | 27,863 | **80.63** | 79,444 | 28.44 | 26,903 | **82.89** | 79,444 | **84.59** | 31,555 | 84.11 |
| MNIST | 32,32 | 79,444 | 96.44 | 36,678 | **96.74** | 79,444 | 26.25 | 26,903 | **96.8** | 79,444 | 96.65 | 36,678 | **96.74** |
| CIFAR-10 | 32,32 | 229,972 | 49.12 | 78,039 | **49.19** | 229,972 | 10.67 | 77,399 | **26.94** | 229,972 | 44.81 | 77,399 | **49.52** |
| CIFAR-100 | 32,32 | 238,792 | 21.86 | 84,611 | **21.97** | 238,792 | 1.09 | 80,459 | **4.08** | 238,792 | 16.4 | 81,099 | **20.87** |

single-hidden-layer setups. Parameter compression remains effective across wavelet types, maintaining 87-91% reduction while preserving accuracy margins.

A cross-architecture analysis reveals consistent in-context learning advantages: 1) Average parameter savings of 89%±4 across all variants; 2) Accuracy improvements of 3.2-14.6 percentage points on small-to-large datasets. These outcomes validate meta-learner's capability to achieve effective balance between parameter reduction and accuracy maintenance across varied basis function implementations (B-spline basis, wavelet and RBF) and architectural depths.

## C.3. Solving PDEs

### C.3.1. EXPERIMENTAL SETUP

We first introduce the exact solution for the PDEs following the (Hu et al., 2024):

$$u_{\text{exact}}(x) = \left(1 - \|x\|_2^2\right) \sum_{i=1}^{d-1} c_i \sin(x_i + \cos(x_{i+1}) + x_{i+1} \cos(x_i)),$$

where $c_i \sim N(0,1)$, and $x \in \mathbb{B}^d$ (the unit ball).

Next, we define the three PDEs.

**Poisson Equation:** The first PDE is the Poisson equation, defined as:

$$\Delta u(x) = g(x), \quad x \in \mathbb{B}^d, \tag{22}$$

where $g(x) = \Delta u_{\text{exact}}(x)$.

**Allen-Cahn Equation:** The second PDE is the Allen-Cahn equation, defined as:

$$\Delta u(x) + u(x) - u(x)^3 = g(x), \quad x \in \mathbb{B}^d, \tag{23}$$

where $g(x) = \Delta u_{\text{exact}}(x) + u_{\text{exact}}(x) - u_{\text{exact}}(x)^3$.

**Sine-Gordon Equation:** The third PDE is the Sine-Gordon equation, defined as:

$$\Delta u(x) + \sin(u(x)) = g(x), \quad x \in \mathbb{B}^d, \tag{24}$$

where $g(x) = \Delta u_{\text{exact}}(x) + \sin(u_{\text{exact}}(x))$.

*Table 8.* Relative $\ell_2$ error and parameters for KANs and MetaKANs at different dimensions.

| DIM | POISSON | | | | ALLEN-CAHN | | | | SINE-GORDON | | | |
| | KAN | | METAKAN | | KAN | | METAKAN | | KAN | | METAKAN | |
| | MSE | PARAM | MSE | PARAM | MSE | PARAM | MSE | PARAM | MSE | PARAM | MSE | PARAM |
|---|---|---|---|---|---|---|---|---|---|---|---|---|
| 20 D | 5.24E-4 | 24,480 | 2.96E-3 | 4,137 | 1.36E-3 | 24,480 | 3.66E-3 | 4,137 | 2.66E-1 | 24,480 | 2.69E-1 | 4,137 |
| 50 D | 2.14E-3 | 33,120 | 4.85E-3 | 5,097 | 2.36E-3 | 33,120 | 6.10E-3 | 5,097 | 6.08E-2 | 33,120 | 5.88E-2 | 5,097 |
| 100 D | 3.77E-3 | 47,520 | 4.59E-3 | 6,697 | 6.53E-3 | 47,520 | 3.37E-3 | 47,520 | 1.91E-2 | 47,520 | 2.60E-2 | 6,697 |

*Table 9.* Ablation study of embedding dimensions and the number of meta-learner (C) on CIFAR-100 dataset.

| | METAKAN | | | | | | | |
| | C=1 | | C=3 | | C=5 | | C=7 | |
| DIM $z$ | ACC | #PARAM | ACC | #PARAM | ACC | #PARAM | ACC | #PARAM |
|---|---|---|---|---|---|---|---|---|
| 1 | 29.90±7.27 | 4,578,565 | 32.03±4.30 | 4,584,215 | 37.75±2.01 | 4,579,305 | 38.00±3.60 | 4,580,731 |
| 2 | 40.40±6.11 | 9,097,397 | 34.58±6.02 | 9,097,799 | 39.34±2.90 | 9,098,585 | 44.17±3.19 | 9,099,371 |
| 4 | 44.13±2.33 | 18,139,285 | 44.37±3.89 | 18,139,751 | 41.85±5.53 | 18,140,665 | 42.04±7.87 | 18,141,579 |

| | METAFASTKAN | | | | | | | |
| | C=1 | | C=3 | | C=5 | | C=7 | |
| DIM $z$ | ACC | #PARAM | ACC | #PARAM | ACC | #PARAM | ACC | #PARAM |
|---|---|---|---|---|---|---|---|---|
| 1 | 40.44±7.68 | 4,575,139 | 37.56±1.48 | 4,575,861 | 46.05±0.41 | 4,576,583 | 44.16±1.01 | 4,576,073 |
| 2 | 38.62±1.93 | 9,096,051 | 40.03±3.46 | 9,096,837 | 46.10±1.85 | 9,097,623 | 49.23±2.23 | 9,098,409 |
| 4 | 42.09±6.33 | 18,137,875 | 42.38±1.53 | 18,138,789 | 49.03±1.52 | 18,139,703 | 52.02±1.07 | 18,140,617 |

These three PDEs are solved using the KAN and MetaKAN models. We apply the exact solution $u_{\text{exact}}(x)$ for the experiments, where the boundary conditions are set to $u = 0$ on $\partial \mathbb{B}^d$.

Both KAN and MetaKAN models have the structure [$n$, 32,32,32,1]. During training, the same number of initial points, boundary points, and interior points are used.

We use a physics-informed loss function based on PINNs, formulated as:

$$\text{loss}_{\text{pde}} = \alpha \text{loss}_{\text{int}} + \text{loss}_{\text{bnd}},$$

where $\text{loss}_{\text{int}}$ represents the residual loss at interior points, which is discretized and evaluated at $N_i$ sampled points. Similarly, $\text{loss}_{\text{bnd}}$ denotes the boundary constraint loss, which is discretized and evaluated at $N_b$ sampled points. The weighting factor $\alpha$ is used to balance the contributions of these two losses in the overall loss function. Following (Zeng et al., 2022), $N_i$ is set to $2000, 4000, 8000, 12000$ for different dimensions $d \in \{20, 50, 100\}$, while $N_b$ is set to 100 points per boundary, resulting in a total of $100d$ boundary points. $\alpha$ is set to 0.01.

### C.3.2. EXPERIMENTAL RESULTS

Table 8 summarizes the performance of the KANs and MetaKANs for solving the Poisson, Allen-Cahn, and Sine-Gordon equations at different dimensions. The table reports the relative $\ell_2$ error and parameter count for various dimensions. These results suggest that MetaKANs can maintain

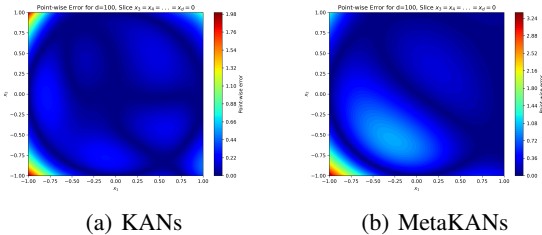

(a) KANs      (b) MetaKANs

*Figure 8.* The relative $\ell_2$ point-wise error visualization for Allen-Cahn equation ($d = 100$).

competitive accuracy while significantly reducing the trainable parameter count. The relative $\ell_2$ error visualization for Allen-Cahn equation ($d = 100$) is shown in Figure 8.

### C.4. Ablation Study

We conduct ablation study on 8-layer MetaKANConv and MetaFastKANConv architectures to systematically analyze the effects of prompt dimensions and meta-learner configuration on CIFAR-100 dataset. Based on channel dimension patterns in different layers, we cluster the 8 layers into $C = 1, 3, 5, 7$ groups for meta-learner assignment with Algorithm 2.

Table 9 demonstrates that increasing dim $z$ generally enhances accuracy, with optimal gains at dim $z$=4 for both architectures. While larger dim $z$ increases parameters (e.g., +13.5M from dim $z$=1 to 4 at $N$=1), the accuracy improvements justify this tradeoff. More significantly, expanding the number of meta-learners ($C$) yields substantial performance gains across both architectures. For MetaKAN,

increasing $C$ from 1 to 3 at dim $z$=4 improves accuracy from 44.13% to 44.37%, while at dim $z$=2, scaling $C$ from 1 to 7 delivers even more pronounced gains (40.40% to 44.17%). The performance progression is particularly remarkable for MetaFastKAN: at dim $z$=4, accuracy increases steadily from 42.09% ($C$=1) to 52.02% ($C$=7), representing a 23.6% relative improvement. This consistent positive correlation between meta-learner count and accuracy highlights the importance of specialized meta-learner for distinct layer clusters.

These results validate our layer clustering strategy in Sec. 3.2.3, demonstrating that task-specific meta-learners ($C > 1$) with sufficient prompt dimensions consistently outperform single meta-learner baselines. The observed performance scaling with $C$ underscores the effectiveness of our grouped meta-learner approach, where dedicated meta-learner for different layer clusters progressively enhance model capacity. MetaFastKAN in particular shows near-linear improvement with increasing $C$, achieving its optimal configuration at $C$=7 and dim $z$=4 with 52.02% accuracy, while MetaKAN reaches peak performance at $C$=3 and dim $z$=4 (44.37%). The minimal parameter overhead (typically ¡0.01%) during this $C$ scaling further confirms the efficiency of our approach.

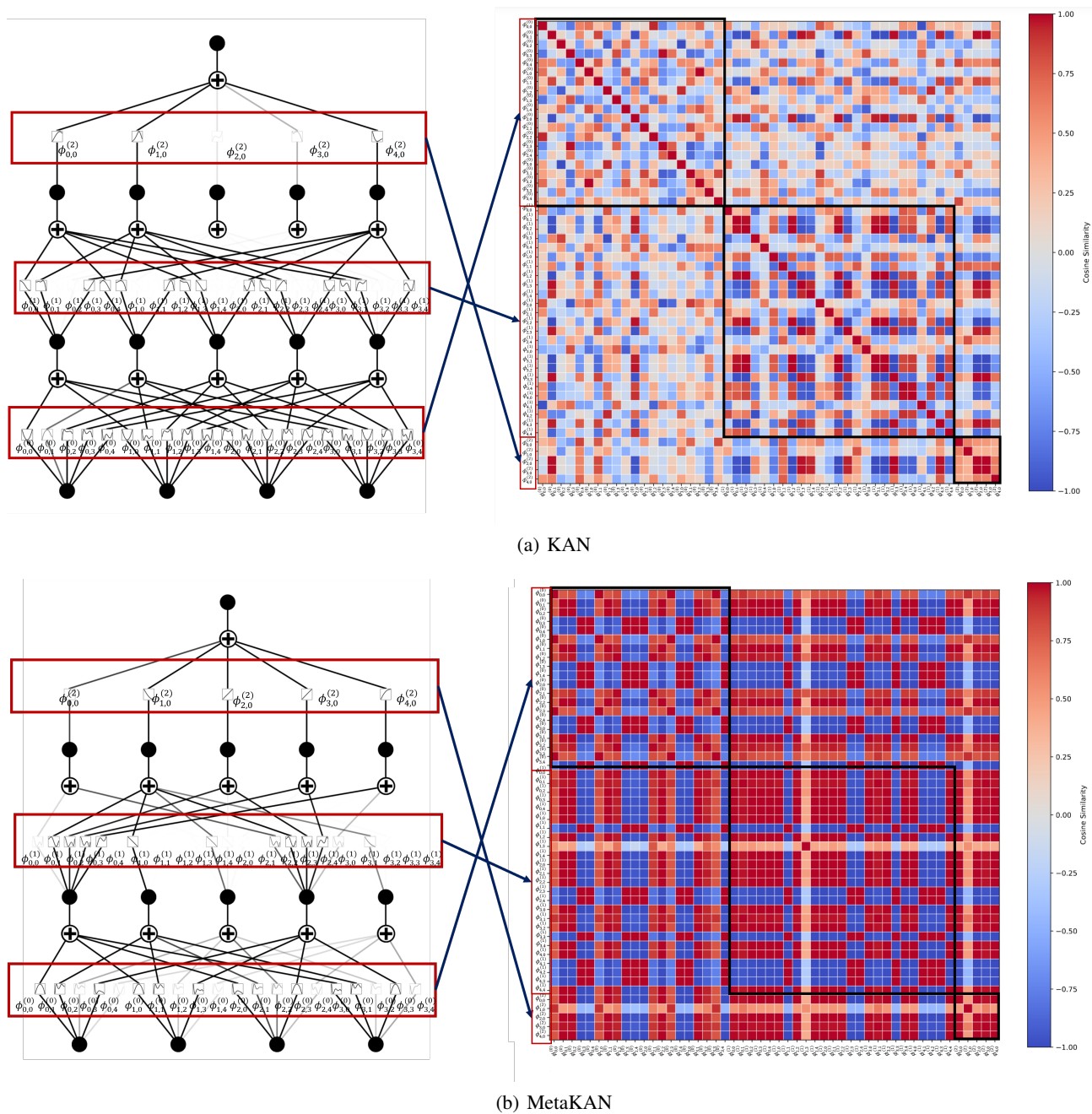

(a) KAN

(b) MetaKAN

*Figure 9.* Comparion of (a) KANs and (b) MetaKANs in terms of learned activation functions (left) and cosine similarity of learned spline coefficient vectors (Right). They are learned for KANs with the structure [4,5,5,1] trained to fit the function $f(\mathbf{x}) = \exp\left(\frac{1}{2}\left(\sin\left(\pi\left(x_1^2 + x_2^2\right)\right) + \sin\left(\pi\left(x_3^2 + x_4^2\right)\right)\right)\right)$. The LHS shows the actual shapes of activation functions learned by KANs and MetaKANs, visualized from the weights of KANs learned by original KANs and predicted by the trained meta-learner of MetaKANs. The RHS displays the cosine similarity between these generated spline coefficient vectors to reveal structural similarities in the learned function shapes. The different color (positive sign or negative sign) can be used to group the activation functions into different classes. We can see that MetaKANs could extract more compact structure of the learned function class spaned by fewer B-spline basis functions, while KANs learn a relatively redundant function class.

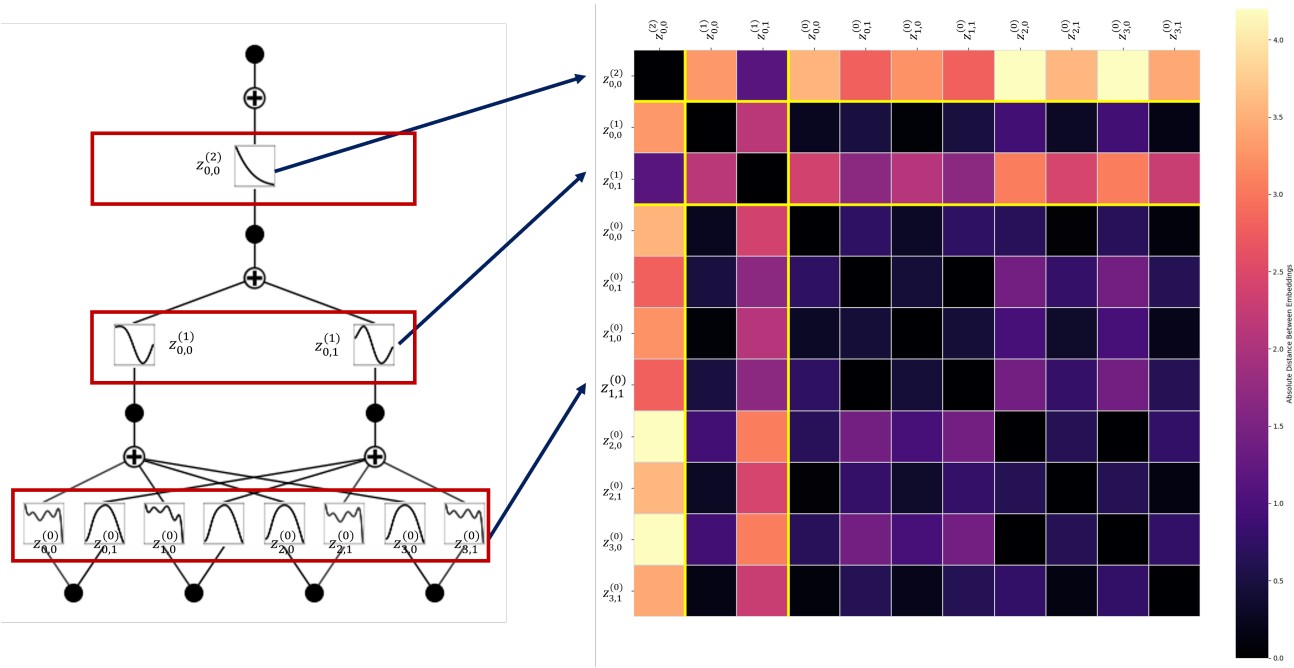

*Figure 10.* Visualization of learned task prompt embedding similarities. Left: Schematic of the hierarchical model structure with illustrative function shapes at different layers. Right: Heatmap depicting pairwise absolute distances between the learned scalar embeddings (labeled $z_\alpha^{(l)}, \alpha \in [n_l] \times [n_{l+1}]$) corresponding to the univariate functions. Darker regions indicate smaller distances (higher similarity in embedding values). KANs fit the function $f(\mathbf{x}) = \exp\left(\frac{1}{2}\left(\sin\left(\pi\left(x_1^2 + x_2^2\right)\right) + \sin\left(\pi\left(x_3^2 + x_4^2\right)\right)\right)\right)$ with structure $[4, 2, 1, 1]$.

