# OpenReview forum: "Improving Memory Efficiency for Training KANs via Meta Learning"
_ICML.cc/2025/Conference — ICML 2025 poster_

### Official Review · Reviewer_rDdN · 2025-02-25

**Overall Recommendation:** 4

**Summary:**

This paper proposes MetaKAN, which leverages a hypernetwork to generate the B-spline coefficients of KANs. Each edge function is associated with a learnable prompt (usually 1D) so the G+K+1 number of coefficients is compressed down to a single scalar, achieving parameter reduction. They apply MetaKANs to function-fitting tasks (Meta + KANs), PDE solving and image cognition tasks (Meta + Convolutional KANs).

## Update after rebuttal
The authors have addressed my questions. Overall this is a great paper, but 5 is a bit of a stretch, so I raised my score from 3 to 4.

**Claims And Evidence:**

The claim about parameter saving (in conclusion) should be tuned down (add "most of the times" or "for large-scale KANs") since results in Figure 2 show that sometimes KANs are more parameter efficient than MetaKANs, especially when KANs themselves are small.

**Essential References Not Discussed:**

None

**Experimental Designs Or Analyses:**

Yes.

**Methods And Evaluation Criteria:**

Yes. They compare models based on model size and performance (loss or accuracy).

**Other Comments Or Suggestions:**

* Boldfaces are not consistently used in tables. For example in Table 2, the comparison on # of parameters for KAN/MetaKAN is not shown for G=5.

**Other Strengths And Weaknesses:**

**Strengths**
* The paper is nicely written and well-motivated
* The paper proposes an interesting and effective solution (probably, see my questions below) to reducing the size of KANs.
* Experiments are diverse -- functional fitting, PDE solving and image recognition.

**Weaknesses**
* Some algorithmic choices seem arbitrary
* In some cases metaKANs appear to have more parameters than KANs. This is not well discussed and acknowledged as a limitation.

**Questions For Authors:**

* I'm not sure why depth (Sec 3.4) makes the story different. Can we simply increase the prompt dimension (usually one) to promote diversity?
* In Table 2, there are a few cases when MetaKANs have more parameters than KANs. why is this the case? Also, boldfaces are not used consistently in Table 2.
* I would love to see this visualization: changing the prompt dimension, how does the B-spline function change? This gives us a sense of what the family of functions looks like.
* Does this meta-learning trick still work when sparsification is added to prune the network?

**Relation To Broader Scientific Literature:**

KANs (Liu et al.) demonstrated the superior performance of KANs over MLPs in small-scale tasks. KANs can become quite parameter-inefficient when they are scaled up. In particular, given the same shape (width and depth), KANs have (G+k+1) times more parameters than MLPs due to the modeling of the B-splines. As a result, it is urgent to reduce the number of parameters for KANs to make them practically useful. This paper presents one interesting solution via meta-learning.

**Theoretical Claims:**

Not applicable. No theoretical claims in the paper.

---

> ### Author Rebuttal · Authors · 2025-04-01
>
> C1: Some algorithmic choices seem arbitrary
>
> A1: We appreciate the reviewer's point regarding the clarity of our algorithmic choices. We acknowledge that the rationale behind certain design choices, such as the specific architecture of the meta-learner (e.g., a two-layer MLP) and the initial dimension of the learnable prompts (e.g., d=1), may not have been sufficiently stated in the original manuscript. In the revised version, we have added explanations for these choices in  Section 3.2.2. We selected a two-layer MLP as the meta-learner because it balances expressive power and parameter efficiency, which is a common choice in many meta-learning and hypernetwork studies [1-3]. For the prompt dimension d=1, we chose it as the most  parameter-minimal  for experimentation. We also explored the impact of different prompt dimensions in the ablation study in  Appendix C.4. The results show that increasing the dimension can improve performance but also comes with additional parameters and potential training instability. This confirms the rationale behind our initial choice of d=1 as the baseline setting, while allowing for flexibility to adjust the dimension when needed. We strive to ensure that all design choices are grounded in empirical observations or inspired by related work, and we will clarify these considerations more explicitly in the revised manuscript.
>
> [1] HyperNetworks, ICLR 2017.
>
> [2] Meta-weight-net, NeurIPS, 2019.
>
> [3] Continual learning with hypernetworks, ICLR, 2020.
>
> C2：Why MetaKANs have more parameters than KANs in Table 2.
>
> A2: We sincerely appreciate the reviewer's careful attention to these important details. You're absolutely correct to note that MetaKANs can have more parameters in certain configurations. Please refer the details about the analysis of the situation to the A3.2 for reviewer tj4z. Besides, the potential benifits of MetaKANs are scaling up to even larger KAN models (the results of large KANs in Table 2; please refer to response for Reviewer A7wa).
>
> And we will reformat all tables to highlight only cases with both MSE improvement and parameter reduction.
>
> C3: Why does depth matter?
>
> A3:
>
> The depth of MetaKAN motivates our design choice of using multiple meta-learners (hypernetworks) in deeper architectures. In deep networks, features become more abstract with depth—early layers capture low-level patterns, while deeper layers model higher-level semantics. Accordingly, the optimal form of activation functions may vary across layers.
>
> Using a single global hypernetwork to generate all activation functions may be overly restrictive, as it must capture diverse functional shape  through input embeddings. Instead, grouping layers and assigning a separate meta-learner to each group allows better specialization, aligning with the hierarchical structure of deep features.
>
> Our ablation study (Appendix C.4) supports this: increasing the prompt dimension from $d=1$ to $d=4$ improves accuracy but also increases variance. Using multiple meta-learners per layer group reduces this variance, especially in deeper networks, suggesting enhanced stability. We plan to further explore the theoretical basis of this effect in future work.
>
> C4: Visualize how B-spline functions vary with prompt dimension
>
> A4: We appreciate the reviewer’s suggestion regarding visualization. In response, we conducted an experiment by varying the promp dimension from 1 to 4 and visualized the corresponding learned B-spline functions  trained on the dataset $f=\exp(\sin(x_1^2+x_2^2)+\sin(x_3^2+x_4^2))$ . These visualizations reveal how increasing the embedding dimension enriches the expressivity of each spline function.
> | Dimension of Embedding | MSE ($10^{-3}$)    |
> |-----------------------|-----------|
> | 1                     | 3.60  |
> | 2                     | 3.45|
> | 3                     | 3.64 |
> | 4                     | 4.21 |
>
>
> With a lower embedding dimension, the splines exhibit relatively simple, smooth transformations. As we increase the embedding dimension to 3 and 4, the spline structures become more complex, reflecting the model’s enhanced capacity to capture more complex functions. For the detailed visualization, please refer to the link https://github.com/icmlrebuttal25/Append
>
> C5: Combining with network pruning
>
> A5: Thank you for your interest in combining our meta-learning approach with network sparsification/pruning. Since MetaKAN focuses on generating activation parameters without altering network structure, pruning can be applied independently. For example, in symbolic regression tasks (Appendix B), we applied sparsification by removing low-activation edges and using symbolic simplification. These results show MetaKAN works well with sparsification techniques.

---

### Official Review · Reviewer_tj4z · 2025-03-14

**Overall Recommendation:** 4

**Summary:**

In this work, the Authors combine KANs with Hypernetworks and show the improved/comparable accuracy at the lower parameter counts.

**Claims And Evidence:**

The claims are supported by the evidence: the proposed model is designed in a standard way, as far as hypernetworks are considered, and is tested on the datasets previously used in the field of KANs.

**Essential References Not Discussed:**

The work thoroughly discusses prior literature on hypernetworks, KANs, and benchmarks.

**Experimental Designs Or Analyses:**

The experimental design is standard for the field of Hypernetworks, therefore, it is validated and justified.

**Methods And Evaluation Criteria:**

Benchmark datasets here are the same as used in prior works in the field of KANs, enabling the direct comparisons of the results here with prior models. Such comparisons are also provided in the paper.

**Other Comments Or Suggestions:**

Line 197: functions -> function

Line 215: formalizes -> formalize

**Other Strengths And Weaknesses:**

Strength: the text is very well written and the experiments are well-documented.

Weakness: the evaluation is only performed on the tasks already solved with KANs (which is a necessary step but may not be a sufficient step). It would be interesting to see MetaKANs solving a task that was not possible to solve with the regular KANs. Such a demonstration would strengthen the work by showing a unique capability of the proposed model.

**Questions For Authors:**

Table 2: There are some apparent inconsistencies. On Line 362, the text says that, at G = 5, MetaKAN achieves a lower MSE in 14 out of 16 functions with parameter reductions. None of these numbers match the table: There are 17 functions, 10 of them correspond to the parameter reduction (by the way, why is that the case?), and all of them are highlighted by the bold font implying the improvement in the MSE. Could you please explain this inconsistency?

Could you please provide an example task that is intractable with KANs but tractable with MetaKAN?

I’m happy to revisit my score based on responses to these questions.

________________________

Post-rebuttal: the typos are fixed and the comments are addressed. Raising my score.

**Relation To Broader Scientific Literature:**

The work extends two lines of research: on Hypernetworks and on KANs, combining them for the first time and showing the utility of such a combination on up-to-date models from literature.

**Theoretical Claims:**

N/A

---

> ### Author Rebuttal · Authors · 2025-04-01
>
> C1: Show a task solvable by MetaKAN but not by KAN
>
> A1:We thank the reviewer for this important suggestion. In fact, our experiments in Section C.1 (High dimensional function) and Table 4 (function fitting task) already demonstrate a key scenario where standard KANs fail while MetaKAN succeeds:
>
> For function $f₂(x) = ∑x² + x³$ at dimension $d=50$, standard KANs completely fail to converge (marked as "NA" in Table 4), while MetaKAN achieves stable performance (MSE=1.15×10⁻³). At $d=1000$, KANs show severe error amplification (MSE=$1.68×10²$), whereas MetaKAN maintains reasonable accuracy (MSE=$1.43×10⁻¹)$ with 7× fewer parameters.
>
> From the theoretical FLOPs analysis in A1 (response to Reviewer A7wa), as network size increases, the computational complexity of KAN grows rapidly with the number of edges. In contrast, MetaKAN benefits from reduced backward-pass complexity, which can result in faster training times on large-scale tasks.
>
> For instance, in the empirical comparison table provided in A1, MetaKAN not only consumes just 30% of the peak memory used by KAN, but also achieves faster training time in the largest network stucture. These results highlight MetaKAN’s strong potential for scaling to even larger models—scenarios where traditional KANs may become impractical due to memory limitations and the high cost of gradient computations.
>
> C2: Line 197: functions -> function Line 215: formalizes -> formalize
>
> A2:We have corrected the typos.
>
> C3: Table 2: There are some apparent inconsistencies. On Line 362, the text says that, at G = 5, MetaKAN achieves a lower MSE in 14 out of 16 functions with parameter reductions. None of these numbers match the table: There are 17 functions, 10 of them correspond to the parameter reduction (by the way, why is that the case?), and all of them are highlighted by the bold font implying the improvement in the MSE. Could you please explain this inconsistency?
>
> A3:We apologize for the confusion in Table 2. The inconsistencies arose because:
>
> 1. The text mistakenly stated "16 functions" (now corrected to 17).
> 2. For detailed comparasion of the parameter count, We take $G=5,k=3$, and the network struncture is $[n_0,n_1,...,n_L]$, the parameter count of KAN is $\sum_{l=0}^{L-1}(n_l\times n_{l+1})\times (G+k+1) =9\sum_{l=0}^{L-1}(n_l\times n_{l+1}) $, and the parameter count of MetaKAN is $\sum_{l=0}^{L-1}(n_l\times n_{l+1})+(d_{hidden}+1)\times (G+k+1) = \sum_{l=0}^{L-1}(n_l\times n_{l+1})+9(d_{hidden}+1)$.The condition for MetaKAN to have fewer parameters than KAN is:  $9\sum_{l=0}^{L-1}(n_l\times n_{l+1})>\sum_{l=0}^{L-1}(n_l\times n_{l+1})+9(d_{hidden}+1)$, which implies the $d_{hidden} < \frac{8}{9}\sum_{l=0}^{L-1}(n_l\times n_{l+1})-1$. So as for some simple network structure (where $\sum_{l=0}^{L-1}(n_l\times n_{l+1})$is small), it does not have the parameter reduction cases.
> 3. Bold font was incorrectly applied to all MSE improvements, even without parameter reduction. We have now highlighted only cases with  both MSE improvement and parameter reduction.

---

> > ### Comment · Reviewer_tj4z · 2025-04-03
> >
> > Thank you for addressing my comments.
> >
> > Specifically, A1 provides a good example that, basically, MetaKAN works at scale, where KAN is intractable.
> >
> > In light of these responses, I am raising my score.

---

### Official Review · Reviewer_A7wa · 2025-03-14

**Overall Recommendation:** 3

**Summary:**

The paper proposes a novel memory optimization method for Kolmogorov-Arnold Networks known as MetaKAN.  MetaKAN leverages the use of meta-learners—2-layer neural networks that generate B-spline coefficients on-the-fly—to reduce the parameter count of KANs to a level comparable to that of MLP’s.  Across various experiments, ranging from simple function fitting to deep learning tasks, MetaKAN consistently achieves an average parameter count reduction relative to KAN ranging from 18% to nearly 90%, while maintaining and often exceeding base KAN performance.

**Claims And Evidence:**

Yes

**Essential References Not Discussed:**

n/a

**Experimental Designs Or Analyses:**

Yes.  I reviewed the experimental designs presented in Section 4 related to function fitting and various tasks using variations on the base convolutional architecture.

**Methods And Evaluation Criteria:**

Yes

**Other Comments Or Suggestions:**

- In Section 3.1.2, Figure 3, the term in the bottom left corner of the matrix uses d_out rather than n_(l+1).  For consistency with the remainder of the figure and the surrounding discussion, I believe this should be n_(l+1).
- In Section 3.2.2, Figure 7, the indexing notation for l is inconsistent with that of Figure 4.  In particular, the union ranges from l=1 to l=L-1, which does not align with Figure 4's range from l=0 to l=L-1.  I believe Figure 7 should be updated to align with Figure 4.
- The parameter counts for MetaKAN provided in Table 1 vs. in Figure 12 are inconsistent.  I believe the Table 1 count is accurate, as it is the parameter count of a 2-layer MLP with dimensions din = 1, dhidden, dout = G + k + 1.

**Other Strengths And Weaknesses:**

Strengths:
- Addresses part of the problematic complexity of KAN by reducing overall parameter count to a level comparable to that of MLP.
- Achieves performance comparable to or exceeding that of base KAN variants on a variety of tasks, including symbolic regression, partial differential equation solving, and image classification.
- Method is portable to several prominent KAN variants, yielding reduced parameter counts and competitive performance.

Weaknesses:
- One of the primary motivations stated in the paper is to reduce the training cost of KAN’s relative to MLP’s; however, the paper does not provide sufficient support for its claim that training costs are reduced.  While a reduction in parameter count may reduce the overall training time, energy demands, etc., increased computational complexity can offset those reductions, and the paper does not provide sufficient evidence to suggest computational complexity or training time are comparable or reduced.
- While the paper identifies in Figure 12 that the theoretical parameter count of MetaKAN is comparable to that of MLP, it does not specify the theoretical parameter count of the modified MetaKAN architecture set forth in Section 3.4 for use in deeper KAN’s.  Given that the parameter count disparity between KAN and MLP is most acute in the deep network context, a discussion of such theoretical complexity is warranted.  While the paper does present empirical evidence of parameter count in several deep network experiments to demonstrate the relative parameter efficiency of MetaKAN in relation to KAN, it does not include comparison results against MLP models.
- The paper does not address in sufficient detail the impact of the MetaKAN innovations on computational complexity / effective computation time relative to MLP, KAN, or any of the KAN variants.  Although the MetaKAN architecture reduces the overall parameter count relative to KAN, it is unclear what the impact of this is on computation time.  One of the primary issues with KAN is its training time relative to MLP, and the paper does not address the impact of MetaKAN on this concern.

**Questions For Authors:**

- What is the computation time of MetaKAN relative to KAN and MLP?  You have demonstrated competitive performance with the reduced parameter count in MetaKAN, but it is unclear what the effect of these changes are on computational complexity.
- Have you performed any experiments testing the performance of MetaKAN relative to MLP (for example, comparing performance with similar architecture dimensions or parameter counts)?  Some discussion on the performance of MetaKAN relative to MLP would be helpful in evaluating its utility in the contexts for which testing has been conducted.

**Relation To Broader Scientific Literature:**

MetaKAN innovates on the original KAN architecture presented in Liu et al., 2024 to address some of its inherent complexity issues.  Various prior approaches have attempted to address these issues by replacing the B-splines utilized in Liu et al., 2024 with various alternatives, such as Chebyshev polynomials (SS et al., 2024), rational functions (Yang & Wang, 2024), wavelets (Bozorgasl & Chen, 2024), and radial functions (Li, 2024).  MetaKAN retains the B-spline functions and reduces overall parameter count by generating B-spline coefficients via a meta-learner (inspired by Kong et al., 2020), rather than training those coefficients directly.

**Theoretical Claims:**

Yes.  I reviewed the justification for the theoretical parameter count of MetaKAN.

---

> ### Author Rebuttal · Authors · 2025-04-01
>
> C1: Complexity comparison and Training time between KAN and MetaKAN.
>
> A1: We thank the reviewer for the insightful question. Below is a summary of theoretical complexity
> 1. Complexity Analysis
> notations see https://github.com/icmlrebuttal25/Append
>
> | Model   | Total FLOPs |
> | ------- | ----------------------------------------- |
> | KAN     | $F_{act} N_{nodes} + N_{edges} C_{spline} + 2N_{node}(K^2 + GK) + 3N_{edges}D_{spline}$ |
> | MetaKAN | $F_{act} N_{nodes} + N_{edges} C_{spline}$ + $N_{edges} C_{meta}$ + $(2N_{edges} + 4d_{hidden}) D_{spline}$ |
>
>
>
> 2.  Advantage condition
>
> MetaKAN becomes cheaper when $N_{edges} > 4d_{hidden}$, typically when $N_{edges} \sim 10^4$ or higher. This is due to trading forward-pass overhead for lower backward complexity. In practice, with >4 layers and >500 neurons per layer ($N_{edges} \sim 10^6$), MetaKAN training time becomes comparable to KAN. For deeper networks (8+ layers, width >1000), MetaKAN can be faster due to backward efficiency gains.
>
> 3. Empirical Comparison
>
> | HiddenWidth x depth| Model   | # Trainable Params | Train Time(s) | Peak Mem.(MB) |
> | ------------ | ------- | ------------------ | ------------- | ------------- |
> | 1024×8       | KAN     | 211M               | 5396          | 7508          |
> | 1024×8       | MetaKAN | 8M                 | 5161          | 2519          |
> | 512×6        | KAN     | 45M                | 1399          | 4108          |
> | 512×6        | MetaKAN | 1.7M               | 1435          | 1212          |
> | 512×4        | KAN     | 31M                | 1006          | 2854          |
> | 512×4        | MetaKAN | 1.2M               | 1060          | 931           |
>
> 4. Potential
>
> These results align well with our theoretical FLOPs analysis. Although MetaKAN introduces a modest overhead in the forward pass due to the hypernetwork, it significantly reduces the cost of the backward pass—a major bottleneck when training large models.  Furthermore, MetaKAN achieves this with drastically fewer trainable parameters and significantly lower memory usage. These advantages indicate that MetaKAN is not only efficient in current setups but also holds strong potential for scaling up to even larger KAN models—scenarios where traditional KANs may become impractical due to memory and gradient computation overhead.
>
>
> C2: Performance and training time comparison between MLP and MetaKAN
> A2:
>
> 1. For deeper structure, the theoretical parameter count of MetaKAN scales mainly with the number of edges and the number of meta-learners leads to a few parameter count increasing.
>
> | Name                            | # Params.                                                    |
> | ------------------------------- | ------------------------------------------------------------ |
> | MLP                             | $N_{edges}+\sum_{l=1}^L n_l$     |
> | KAN                             | $N_{edges} \times (G + k + 1)$   |
> | MetaKAN(Single meta-learner)    | $N_{edges} + (d_{\text{hidden}} + 1) \times (G + k + 1)$ |
> | MetaKAN(Multiple meta-learners) | $N_{edges} + C(d_{\text{hidden}} + 1) \times (G + k + 1)$ |
>
> 2. We compare the two models on two tasks: function fitting and image classification.
>
> On the function fitting task $f_1(x)=\exp\left(\sum_n(1/n)\sin^2(\pi x/2)\right)$ with $n=1000$, MLP trains faster, but its accuracy is significantly lower. MetaKAN retains the inductive bias of spline-based models, achieving both compactness and high accuracy, making it ideal for tasks requiring interpretable and function representations.
>
> For image classification (MNIST), MLPs again train faster and achieve slightly better accuracy. However, MetaKAN still maintains strong performance with substantially fewer parameters.
>
> | Task                 | Model   | Structure          | # Params | Training Time(s) | Metric | Peak Memory(MB) |
> | -------------------- | ------- | ------------------ | -------- | ---------------- | ------ | --------------- |
> | Function Fitting     | MLP     | (1000,3000,3000,1) | 12.7M    | 2.53             | 0.7900 | 193.89          |
> |                      | MetaKAN | (1000,512,512,1)   | 1.6M     | 12.37            | 0.0116 | 372.35          |
> | MNIST Classification | MLP     | (784,16000,10)     | 12.7M    | 282.19           | 97.41% | 195.24          |
> |                      | MetaKAN | (784,2048,10)      | 1.6M     | 717.64           | 97.47% | 2012.32         |
> |                      | MLP     | (784,1800×3,10)    | 7.9M     | 310.55           | 98.11% | 121.33          |
> |                      | MetaKAN | (784,512×3,10)     | 0.93M    | 593.61           | 97.95% | 640.75          |
>
> 3. In summary, while MetaKAN trains slower than MLPs, it retains the core strengths of KANs—strong inductive bias and interpretability—while dramatically reducing parameter count and memory usage compared to KAN. This allows MetaKAN to close the training cost gap with MLPs, offering a practical and scalable alternative that balances efficiency with symbolic inductive biase.

---

> > ### Comment · Reviewer_A7wa · 2025-04-02
> >
> > The additional analysis and results in the rebuttal are very helpful to address my questions. I am raising the score.

---

### Official Review · Reviewer_vpTT · 2025-03-14

**Overall Recommendation:** 4

**Summary:**

This paper concerns a meta-learning approach to training Kolmogorov Arnold Networks (KANs) that enables a reduction in the number of trainable parameters in KANs that is substantially larger than that of standard deep learning models like MLPs. The meta-learning approach is quite standard; it proceeds by assuming that part of the trainable parameters are outputs to a hypernetwork. This output of the hypernetwork then couples together the activation-function-related parameters, significantly reducing the number of parameters (cubic to quadratic). Numerical benchmark illustrates that the thus introduced MetaKAN still achieves good performance while having significantly fewer parameters.

**Claims And Evidence:**

The authors claim that the MetaKAN requires fewer parameters while achieving similar accuracy.

**Essential References Not Discussed:**

More broadly, especially regarding KANs' use in solving PDEs or representing solutions to PDEs, there are other meta-learning approaches that the author can mention. For example, there are well-known papers in the field,

Michael Penwarden, Shandian Zhe, Akil Narayan, Robert M. Kirby, "Physics-Informed Neural Networks (PINNs) for Parameterized PDEs: A Metalearning Approach"

Shaowu Pan, Steven L. Brunton, J. Nathan Kutz, "Neural implicit flow: a mesh-agnostic dimensionality reduction paradigm of spatio-temporal data"

Woojin Cho, Kookjin Lee, Donsub Rim, Noseong Park, "Hypernetwork-based Meta-Learning for Low-Rank Physics-Informed Neural Networks"

**Experimental Designs Or Analyses:**

The experiments are standard, I did not notice any serious issues.

**Methods And Evaluation Criteria:**

The numerical test are standard benchmarks that allow a straightforward comparison with other KANs (INR-type tasks plus standard ML classification tasks). The setup is reasonable and the results look convincing.

**Other Comments Or Suggestions:**

Some suggestions:

- Figure 3, which takes up a significant space, was hard to decipher for this reviewer. Does the diagram on the LHS shows the actual activation functions that were learned? Or are they simply for diagram representation? The cosine similarity on the RHS also appears puzzling... if the MLP pre-output hidden-state is low dimensional, it would already be clear that the output parameters feeding into the activations will also be low-rank. I suggest showing the amplitude. Does the sign in the cosine similarity matter here? If not, perhaps showing its magnitude in log-color scale will be better?

**Other Strengths And Weaknesses:**

Strengths

The paper is well-written and propounds the major points concisely and clearly. The tests seem reasonable.

Weaknesses

In this reviewer's point of view, the paper does not expand sufficiently upon the implications of the specific meta-learning set up. I understand some of this is due to heuristics (ie.. hindsight; some set up works, some not) but are there any settings where the use of MLP arises naturally? For example, Low-Rank PINNs meta-learning approach has some theoretical connections to conservation laws (D. Rim, G. Welper, "Low Rank Neural Representation of Entropy Solutions."

**Questions For Authors:**

- How small can you make the hypernetwork? How does the architecture constrain the weights? What is the extent it can be reduced?

- Is there potential in jointly reducing the coefficients as well as alongside this MetaKAN approach?

- Does the latent parameters Z represent anything physical upon training?

**Relation To Broader Scientific Literature:**

The paper aligns well with recent works in literature attempting to find low-dimensional structures within DL architecture via meta-learning. I believe this is one of the first papers attempting so for KANs, as far as I am aware.

**Theoretical Claims:**

I noticed no substantial theoretical claims, other than parameter counts which appears correct.

---

> ### Author Rebuttal · Authors · 2025-04-01
>
> C1:Why use the MLP as the meta-learner?
>
> A1: We understand the reviewer's interest in the theoretical motivation and deeper implications of employing an MLP as the hypernetwork, particularly in comparison to works like Low-Rank PINNs where architectural choices may have stronger connections to the underlying problem structure (e.g., conservation laws).  We acknowledge that our paper could expand more on this aspect. Our choice of an MLP was primarily driven by several considerations: Firstly, as a universal function approximator, the MLP has been widely adopted in the meta-learning field to learn the common task distribution, such as [1-3] . Secondly, the MLP architecture is relatively simple, computationally efficient, and straightforward to implement and train.
>
> [1] HyperNetworks, ICLR 2017.
>
> [2] Meta-weight-net, NeurIPS, 2019.
>
> [3] Continual learning with hypernetworks, ICLR, 2020.
>
> C2:Clarify Figure 3
>
> A2: We apologize for any lack of clarity in Figure 3. To clarify: the LHS does show the actual shapes of activation functions learned by MetaKAN, visualized from the weights generated by the trained hypernetwork. The RHS displays the cosine similarity between these generated spline coefficient vectors to reveal structural similarities in the learned function shapes. We could utilize the different color (positive sign or negative sign) to group the activation functions into different classes and see the compact structure of the learned function family with only a few function classes.
>
> C3:How small can you make the hypernetwork? How does the architecture constrain the weights? What is the extent it can be reduced?
>
> A3: We thank the reviewer for these insightful questions regarding the hypernetwork's design and its impact. These three aspects – minimum size, architectural constraints, and the extent of reduction – are closely related and central to the MetaKAN approach.
>
> The smallest effective hypernetwork is task-dependent. On simpler tasks, small hidden dimensions (e.g., 4–16) suffice. More complex tasks typically need larger dimensions (e.g., 32–128) to maintain expressivity. The choice reflects finding an point on the efficiency-performance trade-off for the specific application.
>
> The hypernetwork architecture  constrains the generated weights by acting as a shared meta-learner for all activation function coefficients. Instead of learning separate parameters, MetaKAN learns this shared rule  and a unique, low-dimensional embedding Z for each function, as mentioned in our approach. The hypernetwork takes this small embedding Z as input and uses the learned rule to generate the much larger set of spline coefficients (W). Because all coefficients originate from the common function family. The hypernetwork's  hidden dimension restricting the variety and complexity of coefficients that can be produced from the input embedding Z.
>
> The extent of parameter reduction achievable by MetaKAN primarily scales with the ratio of the chosen embedding dimension  to the original spline coefficient dimension, as the total parameters become dominated by the embeddings in larger networks. As illustrated in table, for small networks (e.g., [4,2,2,1]), the hypernetwork's own size , related to hidden dimension significantly impacts the total count, making a compact hypernetwork essential for savings. However, for larger networks, increasing hidden dimension(from 32 to 64) adds negligible parameters (0.1% increase) compared to the vast overall reduction (89%) achieved relative to KAN. Thus, while hypernetwork size matters for small models, the principal reduction factor in practical, larger-scale scenarios is determined by the embedding dimension.
>
> | Structure   | Model                | # Param |
> | ----------- | -------------------- | ------- |
> | [4,2,2,1]   | KAN                  | 117     |
> |             | MetaKAN, d_hidden=4  | 58      |
> |             | MetaKAN, d_hidden=16 | 166     |
> | [784,32,10] | KAN                  | 228,672 |
> |             | MetaKAN, d_hidden=32 | 25,705  |
> |             | MetaKAN, d_hidden=64 | 25,993  |
>
>
> C4:Improvement with KAN coefficient reduction.
>
> A4: We strongly agree that combining MetaKAN with other KAN coefficient reduction techniques holds significant potential. Our generative approach (reducing learned parameters) is largely orthogonal to methods like pruning, quantization, or coefficient sharing  (reducing final coefficient complexity or count). Integrating these strategies could lead to further efficiency gains and is a promising avenue for future research.
>
> C5:Physical meaning of latent parameters Z.
>
> A5: In MetaKAN, the Z embeddings identify the specific activation functions needed per edge. These embeddings organize meaningfully in latent space: nearby Z values produce similar activation shapes. This was verified by comparing similarity matrices of Z and the generated coefficients, which showed strong alignment (as in Figure 3). Thus, Z effectively encodes functional similarity.

---

### Decision · Program_Chairs · 2025-05-01

**Decision:**

Accept (poster)

**Comment:**

This paper introduces MetaKANs, a memory-efficient training framework for Kolmogorov-Arnold Networks (KANs) that replaces direct optimization of activation parameters with a smaller shared meta-learner. The method achieves substantial parameter reduction while maintaining or improving performance across a range of tasks, including symbolic regression, PDE solving, and image classification.

The reviewers were broadly positive. Reviewers vpTT and tj4z praised the clarity of the writing, soundness of the experimental setup, and the novelty of applying meta-learning to KANs. Reviewer A7wa initially raised concerns about the lack of computational complexity analysis and comparisons to MLPs, but was satisfied by the authors' detailed rebuttal and raised their score. Reviewer rDdN similarly appreciated the clarifications and updated their score positively. The paper was generally viewed as well-motivated and technically solid, with some reviewers requesting more extensive ablations or discussion of architectural choices, which were also addressed in the rebuttal.